# ProRL: Effective Reinforcement Learning for Proactive Recommendation via Rectified Policy Gradient Estimation

Hongru Hou [* 1]  Tiehua Mei [* 1]  Denghui Geng [1]  Jinhui Huang [1]
Ao Xu [1]  Hengrui Chen [1]  Jiaqing Liang [1]  Deqing Yang[✉ 1]

## Abstract

Proactive Recommender Systems (PRSs) aim to guide user preference shift toward target items by generating paths of intermediate recommendations. Reinforcement learning (RL) provides a principled framework for optimizing such sequential decision tasks, as path rewards can naturally capture both short-term acceptance and long-term guidance effectiveness. However, naively applying policy gradients to PRS results in deficient gradient estimation. We identify two deficiencies: (1) path-level rewards decompose into step-level rewards with positive mean, creating a *length-dependent bias* that causes gradients to favor path extension over meaningful exploration; (2) weighting each step by the entire path-level reward ignores the decomposition structure, leading to high gradient variance. To rectify these two deficiencies, we propose an effective RL framework **ProRL** with two novel mechanisms for proactive recommendation. First, Stepwise Reward Centering subtracts expected rewards to neutralize length-dependent bias, ensuring that path extension yields zero expected gradient signal. Second, Position-Specific Advantage Estimation leverages the reward decomposition structure to compute step-dependent baselines, reducing gradient variance. Together, these mechanisms yield policy gradients that precisely target path quality. Our experiments on three real-world datasets demonstrate that ProRL significantly outperforms state-of-the-art PRSs. Our code is available at github.com/hongruhou89/ProRL.

---

[*]Equal contribution [1]School of Data Science, Fudan University, Shanghai, China. Correspondence to: Deqing Yang <yangdeqing@fudan.edu.cn>.

*Proceedings of the 43rd International Conference on Machine Learning*, Seoul, South Korea. PMLR 306, 2026. Copyright 2026 by the author(s).

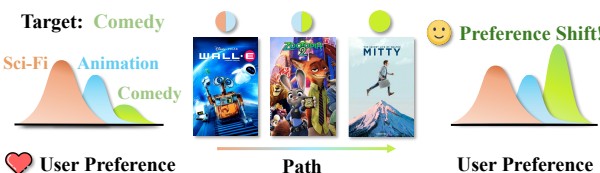

*Figure 1.* A toy example of proactive recommendation. By progressively blending genre (shown via pie charts), each intermediate item in the guidance path maintains the user's engagement while gradually shifting his/her preferences from Sci-Fi to Comedy.

## 1. Introduction

Recommender systems excel at reflecting what users already like (Zhou et al., 2018; Zhai et al., 2024; Hou et al., 2025a; Mei et al., 2025), but platforms are rarely satisfied with merely mirroring past behavior (Liu et al., 2021; Xiang et al., 2025). A streaming service that has just acquired an exclusive jazz catalogue, or an e-commerce site launching a new line of tech accessories, needs users to step beyond their established habits. However, when unfamiliar items are pushed directly into the feed, they are often ignored, lowering acceptance probability (Zheng et al., 2018; Cheng et al., 2016). It exposes a fundamental tension: platforms need certain items to be discovered, while users are anchored in familiar preferences (Li et al., 2019).

This tension motivates a different paradigm of recommendation: rather than abruptly presenting unfamiliar items, a recommender system can gradually shift user preferences toward them through carefully designed paths. **Proactive Recommendation Systems (PRSs)** (Zhu et al., 2023; Lian et al., 2025; Wang et al., 2025b) are then proposed to implement this progressive guidance strategy. Given a user's interaction history and a platform-specified target item that the user has not yet engaged with, a PRS constructs a *path* of intermediate items bridging current user preference to the target item. The system then sequentially recommends items along this path, maintaining acceptance probability at each step while shifting preferences toward the target item. As illustrated in Figure 1, to guide a Sci-Fi fan toward a comedy movie, the system might recommend WALL-E (Sci-Fi + Animation) → Zootopia (Animation + Comedy)

→ The Secret Life of Walter Mitty (Comedy). Each intermediate item remains acceptable to the user, yet the path as a whole cultivates interests for previously unexplored genre.

Designing such paths requires satisfying two objectives simultaneously (Bi et al., 2024). The first is **Path Feasibility**: every intermediate item along the path must achieve high acceptance probability to maintain user engagement. The second is **Guidance Effectiveness**: the complete path must significantly increase the probability that the user eventually accepts the target item. In practice (Zhu et al., 2023; Wang et al., 2025b), these probabilities are estimated by a user simulator, i.e., a recommender system (e.g., SAS-Rec) trained on historical interactions (Section 2). Crucially, these two objectives must be optimized jointly, as locally feasible choices do not guarantee globally effective paths without foresight into their long-term consequences.

Existing PRS research has explored various strategies. Heuristic methods (Bi et al., 2024; Lian et al., 2025) rely on predefined rules to greedily select items at each step, but such local search often yields globally suboptimal paths. LLM-based methods (Wang et al., 2025a;b) plan paths with large language models (LLMs), but are impractical for industrial deployment due to prohibitive costs. Supervised methods (Zhu et al., 2023) treat historical interaction sequences as reference paths which are used to train compact Sequence-to-Sequence models (e.g., T5 (Raffel et al., 2020)). While such lightweight models are attractive for deployment, their reliance on imitating historical data hinders discovering superior paths beyond the training distribution.

In this paper, we employ the lightweight transformer framework of prior work (Zhu et al., 2023), but seek to move beyond imitation of historical interactions. We formalize Path Feasibility and Guidance Effectiveness as quantitative metrics over which proactive recommendation is cast as a reward maximization problem. Reinforcement learning (RL) with policy gradient (Sutton et al., 1999; Mei et al., 2026) handles this problem directly (Section 2.1): the model samples candidate paths, receives reward computed by these metrics, and learns to produce higher-reward paths via gradient-based updates. This exploration-driven paradigm should theoretically enable discovery of effective paths beyond the training distribution. However, preliminary empirical studies (Section 2.2) reveal that standard policy-gradient RL exhibits severe failure modes in PRS.

**Policy Gradient Estimation Deficiencies.** Through empirical studies of applying standard policy-gradient RL to a PRS, we found that it rapidly degenerates into generating *nearly identical overlong paths* (Section 2.2), preventing it from discovering effective, user-specific guidance paths. We trace this failure to two deficiencies in standard policy gradient estimation as below.

*Deficiency 1: Length Shortcut.* We show that path-level rewards in PRS decompose into step-level rewards with a positive mean per step. Thus, longer paths yield higher expected rewards. In standard policy-gradient estimation, variation in sampled path lengths naturally arises, causing length to dominate the gradient signal. This biases the model toward extending paths rather than exploring diverse ones.

*Deficiency 2: High Gradient Variance.* Standard estimation weights each step's gradient by the entire path-level reward. Given the decomposition structure above, this uniform treatment ignores that each step only affects future rewards, resulting in high gradient variance.

**ProRL: Rectified Policy Gradients for PRS.** To address these deficiencies, we propose ProRL, an RL framework that rectifies policy gradient estimation for proactive recommendation. Specifically, *Stepwise Reward Centering* eliminates the length shortcut by subtracting the per-step mean at each position, rectifying the gradient away from spurious length manipulation toward effective path exploration. *Position-Specific Advantage Estimation* reduces gradient variance by exploiting the decomposition structure of path rewards to define a low-variance advantage estimator, rectifying gradient estimates toward their expected values. These two rectifications together yield policy gradient estimates that precisely target path quality, enabling effective optimization of both feasibility and effectiveness.

In summary, the main contributions of this paper include:

1. We identify two gradient estimation deficiencies specific to proactive recommendation, the length shortcut and high gradient variance, that cause standard policy gradients to fail in Proactivate Recommendation System.

2. We propose ProRL, which rectifies these deficiencies through two task-specialized mechanisms. Stepwise Reward Centering adapts classical reward centering to the positive-mean step reward structure of PRS, and Position-Specific Advantage Estimation leverages PRS reward decomposition to compute step-adapted baselines without a learned critic.

3. Extensive experiments on three real-world datasets demonstrate that ProRL significantly outperforms state-of-the-art methods. Ablation studies and cross-evaluator analysis validate each component's contribution and the generalizability of the learned policy.

## 2. Preliminaries

This section formalizes the proactive recommendation task within a reinforcement learning framework (Section 2.1), and then analyzes why standard policy gradient estimation fails in this setting (Section 2.2).

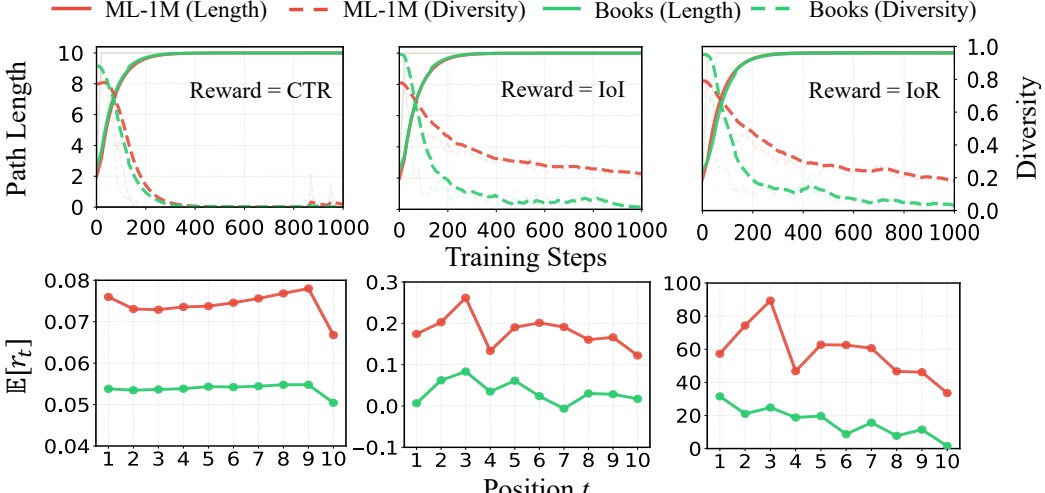

*Figure 2.* Standard policy gradient estimation degenerates into generating nearly identical overlong paths. (**Top row**) Training dynamics under three reward configurations (CTR, IoI, IoR). Each subplot shows path length (solid) and diversity (dashed) for MovieLens-1M (red) and Amazon-Book (green). (**Bottom row**) Expected step-level reward $\mathbb{E}[r_t]$ for each component on MovieLens-1M (red) and Amazon-Book (green). All components exhibit positive mean, enabling the length shortcut.

### 2.1. Basic Framework

As introduced in Section 1, proactive recommendation bridges a user's existing preferences to a platform-specified target item via a path of intermediate recommendations. Formally, given a user's interaction history $S_u$ (sequence of interacted items) and a target item $i_T$, the system generates a recommendation path $L_u = (i_1, \ldots, i_L)$, where $L \leq L_{\max}$.

Following standard practice (Zhu et al., 2023; Bi et al., 2024), we employ a user simulator to estimate acceptance probabilities. The simulator is a recommender model (e.g., SASRec (Kang & McAuley, 2018)) trained on real-world interaction data. It provides estimated probability $P(i \mid S)$ that a user would accept item $i$ given the user's interaction sequence $S$ (representing his/her current preferences). This enables reward computation without online feedback.

Path quality is measured along two dimensions: **Guidance Effectiveness** captures how much the path increases predicted interest in the target, while **Path Feasibility** captures whether users would accept items along the path. To quantify these dimensions, we adopt three standard metrics (Zhu et al., 2023; Bi et al., 2024). Let $\oplus$ denote sequence concatenation and $\mathrm{Rank}(i \mid S)$ denote the ranking position of item $i$ given by the simulator. The metrics are defined as:

$$\mathrm{IoI} := \log P(i_T \mid S_u \oplus L_u) - \log P(i_T \mid S_u),$$
$$\mathrm{IoR} := \mathrm{Rank}(i_T \mid S_u) - \mathrm{Rank}(i_T \mid S_u \oplus L_u),$$
$$\mathrm{CTR} := \frac{1}{|L_u|} \sum_{k=1}^{|L_u|} P\big(i_k \mid S_u \oplus L_u^{<k}\big).$$

Here IoI (Increment of Interest) and IoR (Increment of

Rank) quantify Guidance Effectiveness, while CTR (Click-Through Rate) quantifies Path Feasibility. Effective paths must optimize both dimensions. This naturally motivates a reward defined as a weighted sum of these metrics:

$$R_{\mathrm{path}} = \alpha \cdot \mathrm{IoI} + \beta \cdot \mathrm{IoR} + \gamma \cdot \mathrm{CTR}. \quad (1)$$

With path quality explicitly quantified via the reward in Eq. (1), the goal becomes learning a policy (model) $\pi_\theta(\cdot \mid S_u, i_T)$ that generates high-reward paths. This is naturally framed as an exploration problem: the policy must search a combinatorially large space of candidate paths to discover those with high rewards. RL with policy gradient provides a principled framework for this reward-driven exploration. Specifically, we initialize $\pi_\theta$ with a policy $\pi_0$ pretrained via supervised learning on historical paths (see Appendix E.3 for details). We then update this policy by iteratively sampling paths from $\pi_\theta$ and optimize via policy gradient ascent on the following objective:

$$J(\theta) = \mathbb{E}_{L_u \sim \pi_\theta(\cdot \mid S_u, i_T)}\big[R_{\mathrm{path}}\big] - \lambda \cdot D_{\mathrm{KL}}\big(\pi_\theta \| \pi_0\big). \quad (2)$$

The gradient of $J(\theta)$ consists of two parts: the reward term and the KL term. The KL term can be computed analytically given policy distributions, so we focus on estimating the reward term $\nabla_\theta \mathbb{E}_{\pi_\theta}[R]$. By the policy gradient theorem (Sutton et al., 1999), given $n$ inputs and $m$ sampled paths per input, the standard gradient estimator for $\nabla_\theta \mathbb{E}_{\pi_\theta}[R]$ is:

$$\hat{g}_{\mathrm{std}} = \frac{1}{nm} \sum_{i=1}^{n} \sum_{j=1}^{m} \left[ \sum_{t=1}^{L^{(i,j)}} \nabla_\theta \log \pi_\theta^{(i,j,t)} \cdot R^{(i,j)} \right], \quad (3)$$

where $L^{(i,j)}$ is the path length, $R^{(i,j)}$ is the path reward, and $\pi_\theta^{(i,j,t)}$ denotes the probability. In theory, the policy progres-

sively learns to generate higher-quality paths, moving beyond mere imitation of historical data toward reward-guided discovery. However, as we show next, this standard gradient estimation exhibits severe deficiencies when applied to PRS.

### 2.2. The Length Shortcut

Having established the RL formulation, a natural approach is to directly optimize Eq. (2) with the standard estimator $\hat{g}_{\text{std}}$ (Eq. (3)). However, preliminary experiments reveal that this fails systematically across datasets and reward designs.

**Experimental Setup.** Following Section 2.1, we initialize the policy $\pi_\theta$ with a pretrained model $\pi_0$ and apply standard policy gradient optimization with $L_{\max} = 10$. To isolate the effect of each reward component, we train three separate policies using CTR, IoI, and IoR as the sole reward signal respectively. For each configuration, we repeat the entire pipeline (pretraining + RL) five times and report averaged results. At each training step of RL, we compute two quantities over all rollouts across inputs in the batch: (1) *path length*, the average number of generated items; (2) *path diversity*, item-level Jaccard Similarity among paths.

**Empirical Observation.** Figure 2 (top row) shows the training dynamics on MovieLens-1M and Amazon-Book. Across all reward configurations, we observe a consistent pattern: path length rapidly increases toward the maximum, while path diversity collapses toward nearly zero. Within a few hundred steps, the policy degenerates into generating nearly identical, maximum-length paths for all inputs. This degeneration is common: it occurs regardless of which reward component is used, suggesting a fundamental issue with standard policy gradient estimation in this setting.

**Root Cause: Length-Reward Coupling.** We trace this failure to a structural property of path rewards. We show that any reward function $R$ that maps a path to a scalar value admits a natural decomposition into step-level increments:

$$R(i_1, \ldots, i_L) = \sum_{t=1}^{L} r_t, \tag{4}$$
$$\text{where } r_t := R(i_1, \ldots, i_t) - R(i_1, \ldots, i_{t-1}).$$

This decomposition reveals a critical coupling: if the expected step reward $\mathbb{E}_\pi[r_t]$ [1] is non-zero, then the expected path reward becomes directly dependent on path length.

Figure 2 (bottom row) empirically validates this. We compute $\mathbb{E}_\pi[r_t]$ by averaging step-level rewards across all rollouts collected during the experiments above. Across both datasets and all three reward components, we observe that step-level rewards exhibit *consistently positive mean*. While

---

[1] By expected step reward we actually mean $\mathbb{E}_\pi[r_t | L \geq t]$, since $r_t$ is only defined when the path reaches step $t$. For brevity, we write $\mathbb{E}_\pi[r_t]$ to denote $\mathbb{E}_\pi[r_t | L \geq t]$ when the context is clear.

IoR shows a slow decreasing trend, it remains positive throughout. This positive bias creates a systematic incentive: on average, longer paths yield higher rewards.

One might argue that if longer paths yield higher rewards, the optimal policy should indeed produce long paths. While the global optimum may well correspond to high-quality long paths, the issue lies in the optimization trajectory, not the optimum itself. In early training, the model encounters length variation far more frequently than quality variation among sampled paths. This enables rapid reward improvement through path extension without exploring diverse, high-quality paths. The model thus converges to a local optimum of lengthy but low-quality paths, never reaching the global optimum. Our ablation study (Section 4.3.1) confirms this: removing the length bias yields better final performance with more reasonable path length, demonstrating that the shortcut impedes rather than aids optimization.

**Theoretical Understanding.** Figure 2 (top row) reveals a striking pattern: path length converges to $L_{\max}$ within a few hundred updates, long before the model learns effective item selection. This suggests that early gradients primarily shape the "continue or stop" decision, leaving "which item" to be learned later. To isolate the length mechanism, we consider a simplified model where $\pi_\theta$ stops at each step with a position-independent probability $p = \sigma(\theta)$. The total return $G = \sum_{t=1}^{\tau} r_t$ satisfies $\mathbb{E}[r_t | \tau \geq t] \geq \mu_{\min} > 0$ for all $t$, where $\tau \leq L_{\max}$ is the stopping time.

**Theorem 2.1** (Length Collapse Rate; informal). *Under this setting, let $p(s)$ denote the stop probability under continuous-time gradient flow at training time $s$. Then $p(s) \to 0$ monotonically at rate $O(1/s)$, and the expected path length converges to $L_{\max}$.*

Formal proof is in Appendix A.1. The $O(1/s)$ decay shows that when $\mathbb{E}[r_t | \tau \geq t] \geq \mu_{\min} > 0$, gradient updates systematically reduce stopping probability $p$, making length collapse a structural consequence rather than a tuning artifact. We term this the *length shortcut*.

**Implication.** The analysis suggests a principle for rectifying policy gradient in PRS: *path extension should yield zero expected gain*. Under such condition, the length shortcut disappears and gradients must come from path quality. Section 3 introduces our approach.

## 3. Methodology

### 3.1. Overview

Section 2.2 shows that standard policy gradient estimation fails in PRS due to the *length shortcut*: path-level rewards decompose into step-level rewards with positive mean, causing length to dominate the gradient signal. Beyond this, the decomposition structure suggests an opportunity for

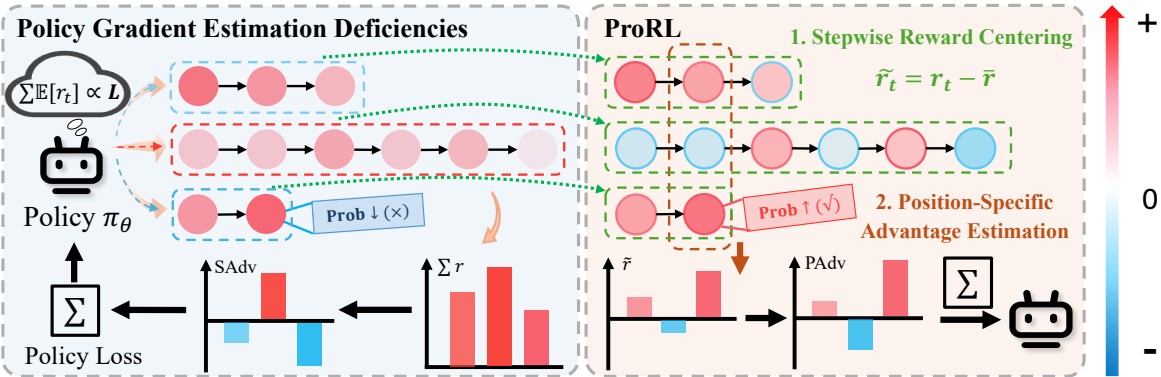

*Figure 3.* Overview of our proposed **ProRL**. (**Left**) Standard policy gradient estimation suffers from the length shortcut. Since step-level rewards accumulate with positive mean ($\sum_{t=1}^{L} \mathbb{E}[r_t] \propto L$), the Sequence-level Advantage (SAdv) and gradient signal are dominated by length variation, causing the model to extend paths rather than explore diverse alternatives. (**Right**) ProRL rectifies gradient estimation through two mechanisms. Stepwise Reward Centering ($\tilde{r}_t = r_t - \bar{r}$) ensures that path extension yields zero expected gain. Position-Specific Advantage Estimation (PAdv) computes step-adapted baselines for effective optimization.

improvement: standard estimation incurs high gradient variance by weighting each step with the entire path reward, which can be reduced through task-specific adaptation to the per-step reward structure. To address both issues, we propose ProRL with the following two mechanisms, which effectively rectify policy gradient estimation.

**Stepwise Reward Centering** (Section 3.2) eliminates the length shortcut: by subtracting the expected reward at each step, we ensure that path extension yields *zero expected gain*, redirecting gradient estimation toward path quality exploration rather than length manipulation.

**Position-Specific Advantage Estimation** (Section 3.3) reduces gradient variance: by computing step-adapted baselines that leverage the decomposition structure of path rewards, we obtain gradient estimates with lower variance.

Together, these rectifications yield policy gradient estimation that achieves effective RL for PRS. Figure 3 illustrates the complete framework.

### 3.2. Stepwise Reward Centering

By Eq. (4), path-level rewards in PRS decompose as $R = \sum_{t=1}^{L} r_t$, where step-level rewards $r_t$ exhibit positive mean $\mathbb{E}_\pi[r_t]$. This couples expected return with path length, causing the length shortcut. Our design is to break this coupling: *path extension should yield zero expected gain*.

We achieve this through reward centering. Empirically, we observe that $\mathbb{E}_\pi[r_t]$ remains relatively stable for many rewards (e.g., IoI; see Figure 2). For simplicity, we use a single global statistic $\bar{r}$ rather than step-specific estimates. We define the centered reward as:

$$\tilde{r}_t = r_t - \bar{r}, \quad \text{where} \quad \bar{r} = \mathbb{E}_\pi[r_*]. \tag{5}$$

Here $\bar{r}$ is the global expected step reward, where the sub-

script "$*$" denotes any step. By construction, $\mathbb{E}_\pi[\tilde{r}_t] = 0$ for all $t$. Therefore, $\mathbb{E}\left[\sum_{t=1}^{L} \tilde{r}_t\right]$ is independent on path length $L$. The length shortcut is eliminated: the model cannot improve rewards by extending paths, and must instead explore deeply into path quality. In practice, we estimate $\bar{r}$ via online accumulation over rollouts of the first training epoch and freeze it for all subsequent epochs. We discuss alternatives to eliminating the length shortcut in Appendix F.5.

**Multi-Objective Reward.** Path quality in PRS involves multiple objectives. To handle this, suppose we have $K$ separate path-level rewards $\{R^{(i)}\}_{i=1}^{K}$, each decomposing into step-level rewards $R^{(i)} = \sum_t r_t^{(i)}$. Since these components have different scales, we extend *centering* to *normalization*:

$$\tilde{r}_t = \sum_{i=1}^{K} w_i \cdot \frac{r_t^{(i)} - \mu^{(i)}}{\sigma^{(i)}}, \tag{6}$$

where $\mu^{(i)} = \mathbb{E}_\pi\left[r_*^{(i)}\right]$ and $\sigma^{(i)} = \sqrt{\mathrm{Var}_\pi\left(r_*^{(i)}\right)}$ are estimated from rollouts during a warm-up epoch, avoiding the drift that would otherwise arise from co-evolving $\mu, \sigma$ and $\pi$ as the policy improves. The resulting normalization centers each component and rescales them to comparable magnitudes, enabling multi-objective optimization.

### 3.3. Position-Specific Advantage Estimation

Stepwise Reward Centering eliminates the length shortcut, but effective training also requires low-variance gradient estimates. Recall from Section 2.1 that the standard gradient estimator $\hat{g}_{\text{std}}$ (Eq. (3)) weights each step's gradient by the total path reward $R^{(i,j)}$. However, the item at step $t$ only affects rewards from $t$ onward; including earlier rewards $r_1, \ldots, r_{t-1}$ introduces irrelevant noise.

We leverage the structural property that path-level rewards decompose into step-level rewards. For step $t$, we define the *reward-to-go* $G_t^{(i,j)} = \sum_{\ell=t}^{L^{(i,j)}} r_\ell^{(i,j)}$ as the cumulative reward from $t$ onward. Replacing $R^{(i,j)}$ with $G_t^{(i,j)}$ excludes past rewards unaffected by the current action:

$$\hat{g}_{\mathrm{rtg}} = \frac{1}{nm} \sum_{i=1}^{n} \sum_{j=1}^{m} \left[ \sum_{t=1}^{L^{(i,j)}} \nabla_\theta \log \pi_\theta^{(i,j,t)} \cdot G_t^{(i,j)} \right]. \quad (7)$$

Variance can be reduced further by centering $G_t$ around its expected value. According to classical RL results (Williams, 1992), subtracting a baseline from the reward-to-go yields an *advantage*, which is an unbiased and lower-variance estimate that measures relative quality rather than absolute return. Traditionally, this requires training an auxiliary critic model, adding complexity and computational cost.

Recent work on LLM alignment, notably GRPO (Shao et al., 2024), avoids the critic by using group Monte Carlo estimation: the baseline is simply the mean path reward across rollouts from the same input, $\bar{R}_i = \frac{1}{m} \sum_{j=1}^{m} R^{(i,j)}$. However, this path-level baseline is shared across all steps, ignoring that the expected reward-to-go varies by position.

Inspired by GRPO, we use the per-step reward structure of PRS to compute *position-specific baseline* $\bar{G}_{i,t}$, the average reward-to-go at step $t$ across all paths from the $i$-th input that reach step $t$. The *position-specific advantage* is then:

$$\bar{G}_{i,t} = \frac{\sum_{j:L^{(i,j)} \geq t} G_t^{(i,j)}}{\sum_{j=1}^{m} \mathbb{I}[L^{(i,j)} \geq t]}, \quad \hat{A}_t^{(i,j)} = G_t^{(i,j)} - \bar{G}_{i,t}. \quad (8)$$

Unlike GRPO's uniform baseline, each step $t$ has its own reference point $\bar{G}_{i,t}$, adapting to the expected future return at that position. Our rectified gradient estimator is:

$$\hat{g}_{\mathrm{rect}} = \frac{1}{nm} \sum_{i=1}^{n} \sum_{j=1}^{m} \left[ \sum_{t=1}^{L^{(i,j)}} \nabla_\theta \log \pi_\theta^{(i,j,t)} \cdot \hat{A}_t^{(i,j)} \right]. \quad (9)$$

This design reduces variance via two well-established mechanisms from classical policy gradient literature (Williams, 1992; Sutton et al., 1999). First, reward-to-go excludes past rewards $r_1, \ldots, r_{t-1}$ that are unaffected by the action at step $t$, removing irrelevant noise from the gradient signal. Second, the position-specific baseline $\bar{G}_{i,t}$ adapts to the expected future return at each position, providing a tighter reference than a path-level baseline. Both techniques are known to preserve unbiasedness while reducing gradient variance (Greensmith et al., 2001). Ablation study (Section 4.3.3) empirically validates the effectiveness of $\hat{g}_{\mathrm{rect}}$.

## 4. Experiments

### 4.1. Experimental Setup

**Datasets.** We conduct experiments on MovieLens-1M (Harper & Konstan, 2015), Steam (Kang & McAuley, 2018), and Amazon-Book (Ni et al., 2019). We construct training data via splitting the raw data by user into training/validation/test sets (8:1:1). Details are in Appendix B.3.

**Baselines.** We compare with four categories of methods, including sequential recommendation method GRU4Rec (Hidasi et al., 2015), BERT4Rec (Sun et al., 2019), LightSANs (Fan et al., 2021), and FEARec (Du et al., 2023); the supervised proactive method IRN (Zhu et al., 2023); heuristic proactive methods IPG (Bi et al., 2024) and ITM-PRec (Lian et al., 2025); LLM-based proactive methods LLM-IPP (Wang et al., 2025a) and T-PRA (Wang et al., 2025b). See Appendix D for details.

**Metrics.** Following prior work (Bi et al., 2024; Wang et al., 2025a;b), we adopt Increment of Interest (IoI) and Increment of Rank (IoR) to measure the guidance effectiveness, and CTR (i.e., HitRate) to measure the path feasibility. Coherence measures the semantic consistency between consecutive items in the path. Details are provided in Appendix C.

**Implementation.** The detailed implementation process and hyperparameters are introduced in Appendix E.

### 4.2. Overall Performance

Table 1 shows that ProRL consistently achieves superior performance across all datasets, outperforming both traditional and state-of-the-art proactive recommendation methods. ProRL achieves the highest guidance effectiveness (IoI and IoR) and path feasibility (CTR and Coherence). Unlike heuristic or LLM-based methods that greedily optimize local objectives and often get trapped in local optima, ProRL directly optimizes the cumulative multi-objective reward over the entire path via rectified policy gradients, achieving both local feasibility and global effectiveness.

Notably, Coherence is not part of the reward function, yet ProRL substantially outperforms all baselines on this unrewarded metric, providing additional evidence that ProRL learns genuinely high-quality paths rather than overfitting to the training reward signal.

A common risk in RL is that the agent exploits specific patterns in the training environment, failing to generalize elsewhere. To verify that ProRL learns a generalized strategy rather than overfitting to the specific reward model (SASRec), we conduct a cross-evaluator analysis. To test this, we use three different recommendation models (GRU4Rec, LightSANs, and BERT4Rec) as unseen evaluators.

As shown in Table 2, the RL policy significantly improves

*Table 1.* Proactive Recommendation performance of all models on different datasets (SASRec as evaluator) in terms of CTR (i.e., HitRate), Coherence, IoI, and IoR. The best performances are highlighted in bold, and the second-best are underlined. The superscript * indicates the Improvement is statistically significant, where the p-value is less than 0.05.

| Dataset | MovieLens-1M | | | | Steam | | | | Amazon-Book | | | |
|---|---|---|---|---|---|---|---|---|---|---|---|---|
| Model | CTR | Coherence | IoI | IoR | CTR | Coherence | IoI | IoR | CTR | Coherence | IoI | IoR |
| GRU4Rec | 0.5143 | 0.3717 | 1.6345 | 77.08 | 0.4312 | 0.7026 | -0.0239 | 15.40 | 0.5544 | 0.5838 | 0.0926 | 83.76 |
| Bert4Rec | 0.5522 | 0.3889 | 1.3402 | 56.03 | 0.4617 | 0.7390 | -0.0055 | 22.00 | 0.5653 | 0.5591 | 0.1042 | 79.34 |
| LightSANs | 0.5211 | 0.3957 | 1.6092 | 85.70 | 0.4215 | 0.7150 | -0.0204 | 21.64 | 0.5626 | 0.5934 | 0.2105 | 148.92 |
| FEARec | 0.5159 | 0.3964 | 1.8770 | 139.85 | 0.4333 | 0.7177 | -0.0216 | 22.93 | 0.5536 | 0.6020 | 0.3671 | 211.36 |
| IRN | 0.8398 | 0.4706 | 1.7277 | 443.29 | 0.3524 | 0.6698 | -0.0481 | 29.66 | 0.4994 | 0.5477 | 0.3111 | 170.73 |
| IPG | 0.4463 | 0.3725 | 2.2537 | 169.28 | 0.2371 | 0.6740 | 0.1758 | 36.26 | 0.5015 | 0.5531 | 1.1067 | 469.68 |
| ITMPRec | 0.4452 | 0.3714 | 2.2719 | 163.80 | 0.2381 | 0.6725 | 0.1804 | 34.50 | 0.5018 | 0.5540 | 1.0980 | 472.50 |
| LLM-IPP | 0.6141 | 0.6288 | 2.4680 | 662.52 | 0.3108 | 0.8022 | 0.0682 | 11.06 | 0.5714 | 0.5132 | 1.6651 | 429.32 |
| T-PRA | 0.4889 | 0.3415 | 2.4867 | 355.16 | 0.2713 | 0.7399 | 0.3339 | 62.04 | 0.5521 | 0.4418 | 1.7261 | 476.93 |
| **ProRL (Ours)** | **0.8543**$^*$ | **0.8422**$^*$ | **2.8504**$^*$ | **728.18**$^*$ | **0.5625**$^*$ | **0.8707**$^*$ | **1.1188**$^*$ | **340.18**$^*$ | **0.8568**$^*$ | **0.6775**$^*$ | **2.9812**$^*$ | **1383.41**$^*$ |

*Table 2.* Cross-evaluator analysis evaluated by the unseen Evaluator GRU4Rec. The best performances are highlighted in bold, and the second-best are underlined. The superscript * indicates the Improvement is statistically significant, where the p-value is less than 0.05.

| Dataset | MovieLens-1M | | | | Steam | | | | Amazon-Book | | | |
|---|---|---|---|---|---|---|---|---|---|---|---|---|
| Model | CTR | Coherence | IoI | IoR | CTR | Coherence | IoI | IoR | CTR | Coherence | IoI | IoR |
| Bert4Rec | 0.6112 | 0.3889 | 2.1632 | 50.18 | 0.5847 | 0.7390 | -0.2425 | 13.46 | 0.5921 | 0.5591 | 0.5643 | 82.85 |
| LightSANs | 0.5771 | 0.3957 | 2.1908 | 67.18 | 0.5616 | 0.7150 | -0.2705 | 12.17 | 0.5817 | 0.5934 | 0.5815 | 124.93 |
| FEARec | 0.5585 | 0.3964 | 2.2509 | 83.90 | 0.5596 | 0.7177 | -0.3271 | 10.89 | 0.5659 | 0.6020 | 0.6102 | 140.87 |
| IRN | 0.7612 | 0.4706 | 2.2012 | 76.12 | 0.4890 | 0.6698 | -0.2773 | 8.59 | 0.5529 | 0.5477 | 0.6637 | 82.36 |
| IPG | 0.4276 | 0.3725 | 2.2409 | 96.24 | 0.3240 | 0.6740 | -0.1084 | 31.44 | 0.5570 | 0.5531 | 0.6524 | 158.01 |
| ITMPRec | 0.4425 | 0.3714 | 2.3068 | 104.35 | 0.3242 | 0.6725 | -0.1044 | 33.93 | 0.5648 | 0.5540 | 0.6733 | 165.25 |
| LLM-IPP | 0.8331 | 0.6288 | 2.3693 | 553.01 | 0.4543 | 0.8022 | -0.0675 | 41.05 | 0.6012 | 0.5132 | 0.7921 | 239.78 |
| T-PRA | 0.4762 | 0.3415 | 2.3167 | 210.24 | 0.4217 | 0.7399 | 0.0523 | 65.98 | 0.6172 | 0.4418 | 1.0762 | 207.89 |
| **ProRL (Ours)** | **0.8460**$^*$ | **0.8422**$^*$ | **2.4560**$^*$ | **649.26**$^*$ | **0.6328**$^*$ | **0.8707**$^*$ | **0.2013**$^*$ | **83.70**$^*$ | **0.8832**$^*$ | **0.6775**$^*$ | **1.7650**$^*$ | **1001.27**$^*$ |

both IoI and IoR. This gain is consistent not only on the original SASRec evaluator, but also on the unseen GRU4Rec evaluator. The consistent gains confirm that the performance boost is not due to overfitting or reward hacking, but rather that ProRL has learned generalizable principles of guidance that transfer across different user behavior models. Full results for additional evaluators are shown in Appendix F.4.

### 4.3. Ablation Study

#### 4.3.1. ABLATION ON RECTIFICATION MODULES

ProRL introduces Stepwise Reward Centering (SRC) and Position-Specific Advantage Estimation (PSAE) to rectify policy gradient estimation. Table 3 presents the ablation results. A notable pattern emerges when SRC is removed (w/o SRC). The CTR on MovieLens-1M and Steam appears unusually high, even exceeding the full ProRL model, but at the cost of severe drops in guidance metrics (IoI, IoR). This anomaly arises because the CTR-based reward has a positive mean. Consequently, without centering, the optimization process is dominated by dense positive click feedback. As a result, the model over-optimizes short-term click probability while failing to capture the sparse, higher-order signals needed for effective guidance. SRC alleviates this bias by enforcing zero expected gain from path extension, thereby balancing optimization across objectives.

*Table 3.* Ablation Studies on ProRL

| Dataset | Model | CTR | IoI | IoR |
|---|---|---|---|---|
| MovieLens-1M | w/o SRC | **0.9731** | 1.2373 | 649.96 |
| | w/o PSAE | 0.7456 | 2.5556 | 695.86 |
| | ProRL | 0.8543 | **2.8504** | **728.18** |
| Steam | w/o SRC | **0.9432** | 0.3217 | 198.30 |
| | w/o PSAE | 0.6311 | 0.7280 | 244.78 |
| | ProRL | 0.5625 | **1.1188** | **340.18** |
| Amazon-Book | w/o SRC | 0.8361 | 1.8825 | 1002.46 |
| | w/o PSAE | 0.8404 | 2.5036 | 1223.78 |
| | ProRL | **0.8568** | **2.9812** | **1383.41** |

#### 4.3.2. ABLATION ON MULTI-REWARD DESIGN

To investigate the contribution of each reward component, we conduct an ablation study across three datasets. As shown in Table 4, the full ProRL model consistently achieves the best performance, validating the necessity of the multi-objective design.

While removing a specific term causes a primary drop in its corresponding metric, we observe degradation across all metrics in certain cases (e.g., w/o IoR on Amazon-Book). This suggests that our reward components are mutually reinforcing and collectively critical for effective policy learning.

*Table 4.* Ablation study on the multi-reward design

| Dataset | Reward | CTR | IoI | IoR |
|---|---|---|---|---|
| MovieLens-1M | w/o Ctr | 0.7722 | 2.1287 | 640.12 |
| | w/o IoI | 0.7875 | 2.2272 | 663.21 |
| | w/o IoR | 0.8377 | 2.6794 | 665.22 |
| | ProRL | **0.8543** | **2.8504** | **728.18** |
| Steam | w/o Ctr | 0.5113 | 1.0126 | 338.28 |
| | w/o IoI | 0.5325 | 0.2856 | 144.49 |
| | w/o IoR | 0.5223 | 0.4540 | 117.46 |
| | ProRL | **0.5625** | **1.1188** | **340.18** |
| Amazon-Book | w/o Ctr | 0.8097 | 2.8217 | 1363.77 |
| | w/o IoI | 0.8359 | 2.3117 | 1261.61 |
| | w/o IoR | 0.7592 | 1.1488 | 125.94 |
| | ProRL | **0.8568** | **2.9812** | **1383.41** |

*Table 5.* Analysis of gradient estimators on ML-1M. We report final performance, average path length at training Epochs 1, 5, 10, and advantage variance (normalized to Epoch 1 of RF).

| Method | Performance | | | Avg. Path Length | | | Advantage Variance | | |
|---|---|---|---|---|---|---|---|---|---|
| | CTR | IoI | IoR | E1 | E5 | E10 | E1 | E2 | E3 |
| RF | 0.581 | 1.626 | 329.8 | 5.2 | 2.9 | 1.5 | 1.00× | 1.18× | 0.94× |
| GRPO | 0.633 | 1.483 | 284.9 | 10.0 | 10.0 | 10.0 | 0.22× | 0.21× | 0.19× |
| A2C | 0.857 | 1.695 | 527.5 | 1.8 | 4.7 | 5.3 | 0.09× | 0.12× | 0.17× |
| RTG | 0.694 | 2.383 | 675.7 | 1.5 | 3.4 | 4.1 | 0.12× | 0.11× | 0.10× |
| **ProRL** | **0.854** | **2.850** | **728.2** | **1.6** | **3.1** | **3.8** | **0.06×** | **0.05×** | **0.05×** |

#### 4.3.3. ABLATION ON GRADIENT ESTIMATORS

To validate the effectiveness of position-specific advantage estimation (Section 3.3), we compare five gradient estimators under identical reward normalization (Eq. (6)), isolating the effect of the advantage method. The estimators are REINFORCE (RF, Eq. (3)), reward-to-go (RTG, Eq. (7)), GRPO (path-level baseline), A2C (Mnih et al. (2016), see Appendix E.3 for details), and ProRL (Eq. (9)).

Figure 4 shows training dynamics, where all curves report test metrics. ProRL achieves the best overall performance with steady improvement across all metrics. Table 5 further jointly analyzes final performance, path length stability during training, and gradient variance on ML-1M. ProRL achieves the highest guidance metrics, substantially surpassing both GRPO and A2C in guidance effectiveness. The path length and variance columns in Table 5 reveal the mechanism behind these performance differences. We track average path lengths at training Epochs 1, 5, and 10. RF and GRPO exhibit opposite failure modes, with RF collapsing to length 1.5 while GRPO saturates at $L_{max} = 10$ throughout training. A2C shows moderate but unstable growth. In contrast, ProRL and RTG consistently converge to stable, moderate lengths around 3 to 4 steps. This stability is directly tied to gradient variance. ProRL achieves the lowest variance (∼5% of RF at Epoch 1), and RTG also maintains low variance, corroborating both methods' length stability.

*Table 6.* Experimental Results with Different Training Stages

| Dataset | Model | CTR | IoI | IoR |
|---|---|---|---|---|
| MovieLens-1M | Pretrain | **0.8671** | 0.8600 | 254.43 |
| | RL | 0.8543 | **2.8504** | **728.18** |
| Steam | Pretrain | **0.7453** | 0.4230 | 101.16 |
| | RL | 0.5625 | **1.1188** | **340.18** |
| Amazon-Book | Pretrain | 0.6410 | 0.1650 | 72.92 |
| | RL | **0.8568** | **2.9812** | **1383.41** |

*Table 7.* Capacity analysis of pretrained model via Rollout@K.

| Dataset | Max-IoI | | | Max-IoR | | |
|---|---|---|---|---|---|---|
| | @1 | @5 | @10 | @1 | @5 | @10 |
| MovieLens-1M | 1.1347 | 2.7779 | 3.3585 | 294.53 | 717.69 | 851.03 |
| Steam | 0.2395 | 1.8728 | 2.4803 | 57.89 | 818.11 | 1074.35 |
| Books | 0.1523 | 2.2524 | 3.0780 | 52.47 | 1132.01 | 1509.70 |

A key finding is that A2C's variance *increases* over training (0.09× → 0.17×), as its learned critic fails to track the evolving policy and produces progressively noisier baselines. ProRL's analytic baseline (Eq. (8)), computed directly from rollout statistics, adapts naturally without this drift.

### 4.4. Quantitative Analysis of Training Stages

To understand the evolution from pretraining to RL, we conduct a two-stage analysis. We first evaluate performance at each stage, then investigate the mechanism behind improvement by probing the pretrained model's latent capacity.

**The Leap from Feasibility to Effectiveness.** As shown in Table 6, the pretrained model achieves high CTR, establishing a foundation for path feasibility. However, its guidance effectiveness remains limited. The RL stage breaks this bottleneck. By shifting the objective from likelihood maximization to cumulative reward maximization, RL improves guidance effectiveness while maintaining path feasibility. This confirms that rectified policy gradients (SRC + PSAE) enable effective RL optimization that discovers high-quality paths beyond the pretraining distribution.

**Mechanisms for Eliciting Pre-existing Capabilities.** The dramatic gain in effectiveness raises a question: *Does RL impart new capabilities, or unlock potential already existing in the pretrained model?*

To answer this, we probe the latent capacity of the fixed pretrained model using **Rollout@K** analysis. We sample $K$ paths for each input from the pretrained model and record the maximum IoI/IoR achieved. As shown in Table 7, while greedy generation (@1) is weak, the latent potential (@10) is remarkably high, often matching RL's final performance. This reveals that our RL stage actually functions as a **probabilistic rectifier**, identifying high-quality guidance paths

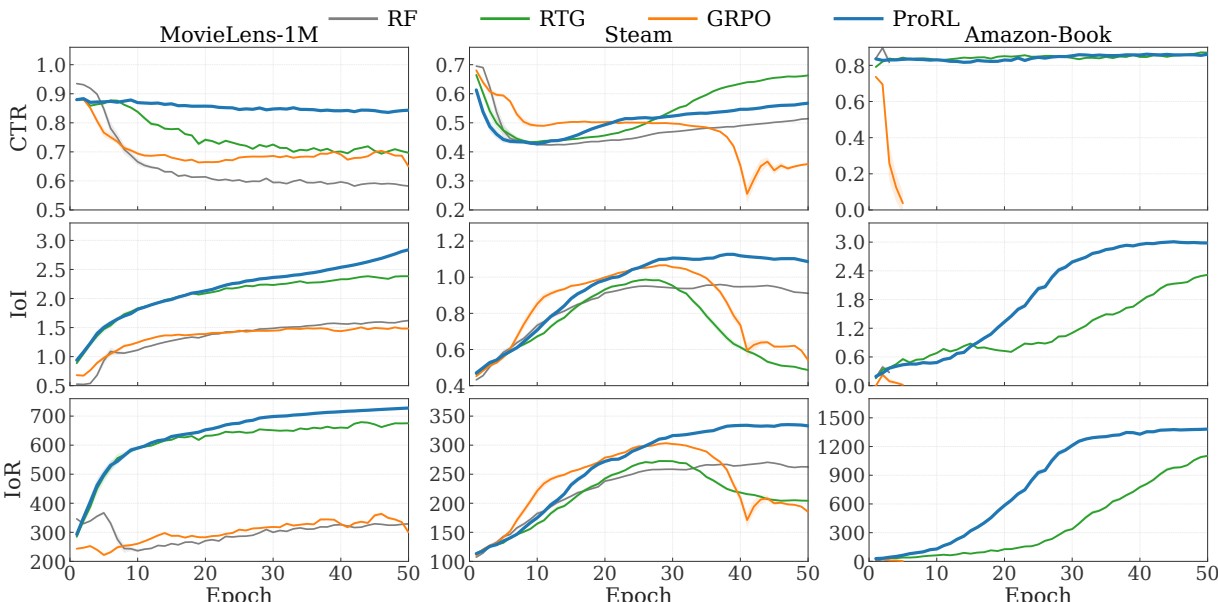

*Figure 4.* Training dynamics of gradient estimators on MovieLens-1M, Steam, and Amazon-Book. All methods use identical multi-objective rewards with the same weights; only the gradient estimator differs. RF: REINFORCE (Eq. (3)); RTG: reward-to-go (Eq. (7)); GRPO: REINFORCE with path-level baseline; ProRL: position-specific advantage (Eq. (9)). ProRL achieves the best balance between path feasibility (CTR) and guidance effectiveness (IoI, IoR), while RTG sacrifices guidance metrics for higher CTR.

in the low-probability tail of the pretrained distribution and redistributing probability mass towards them.

## 5. Related Work

**Sequential Recommendation.** Sequential recommendation models user history to predict future behaviors. GRU4Rec (Hidasi et al., 2015) pioneered RNNs for temporal modeling. SASRec (Kang & McAuley, 2018) adapts self-attention with causal masking, whereas BERT4Rec (Sun et al., 2019) employs bidirectional objectives to deepen context understanding. Recent advances optimize this backbone. LightSANs (Fan et al., 2021) introduces low-rank decomposed attention for linear scalability, and FEARec (Du et al., 2023) leverages frequency domain learning for multi-scale information. However, these methods focus on fitting historical preferences and fail to shift user preferences.

**Proactive Recommendation.** Proactive recommendations aim to shift user preferences toward target items. IRN (Zhu et al., 2023) introduces a Transformer-based supervised method with a Personalized Impressionability Mask to model user receptiveness. IPG (Bi et al., 2024) selects intermediate items by jointly evaluating local feasibility and guidance effectiveness via predefined heuristics, while ITM-PRec (Lian et al., 2025) further incorporates intention-level features for finer-grained characterization. More recently, LLM-IPP (Wang et al., 2025a) leverages LLMs via Chain-of-Thought for path planning, and T-PRA (Wang et al.,

2025b) employs an LLM-based Actor-Critic framework. However, supervised methods cannot explore paths beyond historical data; heuristic methods greedily optimize local objectives and often yield suboptimal paths; and LLM-based methods incur prohibitive deployment costs.

## 6. Conclusion

We present ProRL, a reinforcement learning framework for proactive recommendation via rectified policy gradient estimation. Our analysis reveals two deficiencies in standard policy gradient estimation for PRS: the *length shortcut* and *high gradient variance*. To address these issues, we introduce two rectifications. Stepwise Reward Centering eliminates the length shortcut by ensuring path extension yields zero expected gain, while Position-Specific Advantage Estimation reduces variance by exploiting reward decomposition. Together, these rectifications yield policy gradients that align with target path quality, enabling effective optimization of both feasibility and effectiveness. Experiments on three real-world datasets confirm that ProRL significantly outperforms state-of-the-art methods, and cross-evaluator analysis validates that the learned guidance strategy generalizes beyond the training reward model.

## Acknowledgment

This work was supported by the Chinese NSF General Program (No.62572129).

## Impact Statement

This paper contributes to the field of proactive recommendation by introducing a reinforcement learning-based guidance framework. Our work aims to enhance the capability of recommender systems to proactively assist users in exploring new interests or achieving specific goals. While our method focuses on optimizing guidance efficiency, we acknowledge the importance of aligning such proactive strategies with user utility and ethical standards to ensure a positive user experience.

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

# A. Theoretical Analysis

This section provides the formal statement and complete proof of Theorem 2.1, which is informally presented in Section 2.2 of the main text.

## A.1. Length Collapse: Formal Statement and Proof of Theorem 2.1

Section 2.2 presents an informal statement of Theorem 2.1, which characterizes how gradient updates systematically reduce stopping probability when step-level rewards have positive mean. Here we provide the formal statement and complete proof.

**Setup.** To isolate the length mechanism from item selection, we consider a simplified model where the policy makes only `continue` or `stop` decisions. At each step $t \in \{1, \dots, L_{\max}\}$, the policy stops with probability $p = \sigma(\theta) \in (0, 1)$, where $\sigma(\cdot)$ is the sigmoid function and $\theta \in \mathbb{R}$ is the learnable parameter. Let $\tau \leq L_{\max}$ denote the stopping time, and define the total return as

$$G = \sum_{t=1}^{\tau} r_t, \tag{10}$$

where $r_t$ is the step-level reward at time $t$. The objective is $J(\theta) = \mathbb{E}_{\pi_\theta}[G]$. For this stylized analysis, we treat the conditional means $\mathbb{E}[r_t \mid \tau \geq t]$ as fixed (i.e., not depending on $\theta$), so that $J(\theta)$ can be analyzed as a function of $p = \sigma(\theta)$.

**Theorem A.1** (Length Collapse Rate). *Suppose the expected step-level reward satisfies $\mathbb{E}[r_t \mid \tau \geq t] \geq \mu_{\min} > 0$ for all $t \in \{1, \dots, L_{\max}\}$. Consider gradient flow dynamics*

$$\frac{\mathrm{d}\theta(s)}{\mathrm{d}s} = \frac{\mathrm{d}J}{\mathrm{d}\theta}\big(\theta(s)\big), \qquad \theta(0) = \theta_0, \tag{11}$$

*and let $p(s) = \sigma(\theta(s))$. Then:*

1. *$p(s)$ is strictly decreasing and $p(s) \to 0$ as $s \to \infty$;*

2. *There exist constants $S, K > 0$ such that for all $s \geq S$,*

$$p(s) \leq \frac{K}{s}. \tag{12}$$

*Consequently, the expected path length $\mathbb{E}[\tau] \to L_{\max}$ as $s \to \infty$.*

*Proof.* We proceed in four steps.

**Step 1: Express $J$ as a function of $p$.** Define the event

$$E_t := \{\tau \geq t\} = \{a_1 = \dots = a_{t-1} = \texttt{cont}\}, \tag{13}$$

representing that the policy has not stopped before step $t$. Under the homogeneous stopping policy, $\mathbb{P}(E_t) = (1 - p)^{t-1}$. By the tower property,

$$J = \sum_{t=1}^{L_{\max}} \mathbb{E}[r_t \mathbf{1}_{E_t}] = \sum_{t=1}^{L_{\max}} \mathbb{P}(E_t)\,\mathbb{E}[r_t \mid E_t] = \sum_{t=1}^{L_{\max}} \mu_t (1 - p)^{t-1}, \tag{14}$$

where $\mu_t := \mathbb{E}[r_t \mid E_t] \geq \mu_{\min} > 0$ by assumption. Under the setup above, $\{\mu_t\}$ are treated as constants, so $J$ can be viewed as a function of $p$.

**Step 2: Show $\mathrm{d}J/\mathrm{d}\theta < 0$.** Differentiating with respect to $p$:

$$\frac{\mathrm{d}J}{\mathrm{d}p} = -\sum_{t=2}^{L_{\max}} (t - 1)\mu_t (1 - p)^{t-2} \leq -\mu_{\min} \sum_{t=2}^{L_{\max}} (t - 1)(1 - p)^{t-2} < 0 \tag{15}$$

for $p \in (0, 1)$. Since

$$\frac{\mathrm{d}p}{\mathrm{d}\theta} = p(1 - p) > 0, \tag{16}$$

the chain rule gives

$$\frac{\mathrm{d}J}{\mathrm{d}\theta} = \frac{\mathrm{d}J}{\mathrm{d}p} \cdot \frac{\mathrm{d}p}{\mathrm{d}\theta} < 0. \tag{17}$$

Under gradient ascent flow, $\frac{\mathrm{d}\theta(s)}{\mathrm{d}s} = \frac{\mathrm{d}J}{\mathrm{d}\theta} < 0$, so $\theta(s)$ is strictly decreasing, and consequently $p(s) = \sigma(\theta(s))$ is strictly decreasing.

**Step 3: Prove $p(s) \to 0$.** Since $p(s) \in (0,1)$ is monotonically decreasing, the limit $p_\infty := \lim_{s\to\infty} p(s)$ exists with $p_\infty \in [0,1)$. Suppose for contradiction that $p_\infty > 0$.

Because $p(s) \to p_\infty$, there exists $S_1$ such that for all $s \geq S_1$,

$$|p(s) - p_\infty| \leq \min\left\{\frac{p_\infty}{2}, \frac{1-p_\infty}{2}\right\}. \tag{18}$$

In particular, for all $s \geq S_1$ we have

$$p(s) \geq \frac{p_\infty}{2} \qquad \text{and} \qquad 1 - p(s) \geq \frac{1-p_\infty}{2}, \tag{19}$$

hence

$$p(s)\big(1 - p(s)\big) \geq \frac{p_\infty}{2} \cdot \frac{1-p_\infty}{2}. \tag{20}$$

Note that for all $p \in (0,1)$,

$$\frac{\mathrm{d}J}{\mathrm{d}p} = -\sum_{t=2}^{L_{\max}} (t-1)\mu_t (1-p)^{t-2} \leq -\mu_2 \leq -\mu_{\min}. \tag{21}$$

Thus, for $s \geq S_1$,

$$\frac{\mathrm{d}\theta(s)}{\mathrm{d}s} = \frac{\mathrm{d}J}{\mathrm{d}\theta} = \frac{\mathrm{d}J}{\mathrm{d}p}\big(p(s)\big) \cdot \frac{\mathrm{d}p}{\mathrm{d}\theta}\big(\theta(s)\big) \leq -\mu_{\min} \cdot \frac{p_\infty}{2} \cdot \frac{1-p_\infty}{2} =: -c < 0. \tag{22}$$

This implies $\theta(s) \to -\infty$ as $s \to \infty$, hence $p(s) = \sigma(\theta(s)) \to 0$, contradicting $p_\infty > 0$. Therefore $p_\infty = 0$.

**Step 4: Establish the $O(1/s)$ convergence rate.** By the chain rule,

$$\frac{\mathrm{d}p(s)}{\mathrm{d}s} = \frac{\mathrm{d}p}{\mathrm{d}\theta} \cdot \frac{\mathrm{d}\theta}{\mathrm{d}s} = \frac{\mathrm{d}p}{\mathrm{d}\theta} \cdot \frac{\mathrm{d}J}{\mathrm{d}\theta} = \frac{\mathrm{d}p}{\mathrm{d}\theta} \cdot \frac{\mathrm{d}J}{\mathrm{d}p} \cdot \frac{\mathrm{d}p}{\mathrm{d}\theta} = p(s)^2(1-p(s))^2 \cdot \frac{\mathrm{d}J}{\mathrm{d}p}\big(p(s)\big). \tag{23}$$

Using $\frac{\mathrm{d}J}{\mathrm{d}p} \leq -\mu_{\min}$,

$$\frac{\mathrm{d}p(s)}{\mathrm{d}s} \leq -\mu_{\min}\, p(s)^2(1-p(s))^2. \tag{24}$$

Since $p(s) \to 0$, there exists $S_0$ such that $p(s) \leq 1/2$ for all $s \geq S_0$, giving $(1-p(s))^2 \geq 1/4$. Thus, for $s \geq S_0$,

$$\frac{\mathrm{d}p(s)}{\mathrm{d}s} \leq -\frac{\mu_{\min}}{4}\, p(s)^2. \tag{25}$$

Define $q(s) := 1/p(s)$. Then

$$\frac{\mathrm{d}q(s)}{\mathrm{d}s} = -\frac{1}{p(s)^2} \frac{\mathrm{d}p(s)}{\mathrm{d}s} \geq \frac{\mu_{\min}}{4}. \tag{26}$$

Integrating from $S_0$ to $s$ yields

$$\frac{1}{p(s)} \geq \frac{1}{p(S_0)} + \frac{\mu_{\min}}{4}(s - S_0), \tag{27}$$

which implies

$$p(s) \leq \frac{4}{\mu_{\min}(s - S_0)}. \tag{28}$$

Setting $S = S_0 + 1$ and $K = 4S/\mu_{\min}$, we obtain $p(s) \leq K/s$ for all $s \geq S$. $\qquad\square$

**Implication.** The $O(1/s)$ decay rate demonstrates that length collapse is a structural consequence of positive step-level reward means, not a tuning artifact. Under standard policy gradient updates, the stopping probability vanishes at a polynomial rate, causing path length to converge to $L_{\max}$ regardless of path quality. This motivates Stepwise Reward Centering (Section 3.2), which enforces zero-mean stepwise gains and thereby avoids length collapse.

## B. Data Construction and Implementation

This section describes the data construction pipeline and implementation details of ProRL. We first clarify how our implementation relates to the item-level formulation (Section B.1). We then present the dataset statistics (Section B.2), the training data construction process (Section B.3), and the semantic tokenization procedure (Section B.4).

### B.1. Implementation via Semantic IDs

The item-level formulation in Section 2.1 provides a conceptual framework where each action selects an item $i \in \mathcal{I}$. In practice, we instantiate this framework using **semantic IDs**, a widely adopted technique in generative recommendation (Zhai et al., 2024; Hou et al., 2025b). This subsection clarifies the relationship between the conceptual formulation and our implementation, and establishes their theoretical compatibility.

**Semantic ID Representation.** Each item $i$ is represented as a sequence of $K$ discrete tokens $(c_1^i, c_2^i, \ldots, c_K^i)$ via Residual Quantized VAE (detailed in Section B.4). In our experiments, $K = 4$. The policy $\pi_\theta$ autoregressively generates these tokens, producing a sequence of $K \cdot L + 1$ tokens (including the EOS token) that decodes to a path of $L$ items.

**Theoretical Compatibility.** The semantic ID implementation is fully compatible with the item-level framework presented in the main text:

- When $K = 1$, the two formulations are identical.

- When $K > 1$, generating an item $(c_1, \ldots, c_K)$ can be viewed as a single composite action in the item-level formulation.

Formally, let $\tilde{\pi}_\theta(i \mid s)$ denote the probability of generating item $i$ under the semantic ID policy. This probability is given by:

$$\tilde{\pi}_\theta(i \mid s) = \prod_{j=1}^{K} \pi_\theta(c_j^i \mid s, c_1^i, \ldots, c_{j-1}^i). \tag{29}$$

The item-level reward $r_t$ and advantage $\hat{A}_t$ (Eq. (5) and Eq. (8)) are computed at the item level after decoding. During backpropagation, the gradient is distributed to all $K$ tokens of item $i_t$:

$$\nabla_\theta \log \tilde{\pi}_\theta(i_t \mid s_t) \cdot \hat{A}_t = \sum_{j=1}^{K} \nabla_\theta \log \pi_\theta(c_j^{i_t} \mid \cdot) \cdot \hat{A}_t. \tag{30}$$

That is, **all tokens within the same item share the item-level advantage**, preserving the semantics of our rectified policy gradient estimator (Eq. (9)).

**Benefits of Semantic IDs.** This implementation offers practical advantages:

1. *Reduced action space*: The vocabulary size reduces from $|\mathcal{I}|$ items to $|\mathcal{C}|$ codebook entries, where typically $|\mathcal{C}| \ll |\mathcal{I}|$.

2. *Semantic generalization*: Items with similar semantics share token prefixes, enabling better generalization.

3. *Compatibility with Sequence-to-Sequence*: Standard encoder-decoder Transformers naturally handle token sequences.

Crucially, the theoretical analysis in Section 2.2 and Appendix A.1 remains valid: it depends only on the decomposition $R = \sum_t r_t$ and the property $\mathbb{E}[r_t] > 0$, which hold at the item level regardless of how items are tokenized.

*Table 8.* Dataset statistics.

| Dataset | # Users | # Items | # Interaction | # Avg. Int. |
|---|---|---|---|---|
| MovieLens-1M | 6,040 | 3,040 | 1,000,209 | 165.59 |
| Steam | 2,567,538 | 15,474 | 7,793,069 | 3.03 |
| Amazon-Book | 10,297,355 | 4,493,336 | 29,475,453 | 2.86 |

*Table 9.* Processed Dataset statistics.

| Dataset | # Training | # Validation | # Test |
|---|---|---|---|
| MovieLens-1M | 56,141 | 6,771 | 6,177 |
| Steam | 78,152 | 9,522 | 9,956 |
| Amazon-Book | 66,881 | 8,986 | 7,921 |

### B.2. Dataset Statistics

Our experiments utilize three public datasets: MovieLens-1M[2], Steam[3], and Amazon-Book[4]. The MovieLens-1M dataset includes a total of 1,000,209 interactions, with an average interaction length of 165.59 per user, and a total of 3,040 items. The Steam dataset consists of 7,793,069 interactions, with an average length of 3.03 per user, and a total of 15,474 items. The Amazon-Book dataset consists of 29,475,453 interactions, with an average length of 2.86 per user, and a total of 4,493,336 items.

We apply k-core preprocessing to filter out users and items with fewer than a particular interactions to ensure sufficient data for model training. Specifically, for the MovieLens-1M and Steam datasets, we apply 20- and 40-core filters to users and items, respectively. For the Amazon-Book datasets, we conduct a 100-core filter for users and a 40-core filter for items.

**Coherence Knowledge Exploitation.** Before getting the Smooth-Guided Data, we need to pre-define the attributes used to exploit the coherence between adjacent items. For the MovieLens-1M datasets, we regard the genres of movies as bridge attributes, which means the adjacent movies that share at least one genre are correlated. We filter out the "Drama" genre, as "Drama" is a large category showing the least correlation. For the Steam dataset, we regard the categories, publisher, and developer as the bridge attributes. For the Amazon-Book dataset, we use the category as the bridge attribute to exploit subsequences with adjacent correlation.

**Data Splitting.** To ensure a robust evaluation and prevent information leakage across users, we adopt a user-centric data partitioning strategy. Specifically, the entire pool of unique users, along with their corresponding interaction subsequences, is randomly partitioned into training, validation, and testing sets with a ratio of 80%, 10%, and 10%, respectively. This split ensures that the model is tested on previously unseen users, thereby evaluating its generalization capability in proactive scenarios. Detailed statistics for the processed datasets, including the number of proactive logs comprising: history interaction and the target item, are summarized in Table 9.

### B.3. Smooth Guided Data Construction

While pre-trained Language Models have shown promise (Brown et al., 2020; Ouyang et al., 2022), directly adapting their weights to recommendation often incurs *negative transfer* due to the semantic gap between linguistic contexts and behavioral patterns (Zhang et al., 2025a; Bao et al., 2023; Wang et al., 2025c). To avoid this noise, we opt to pre-train our proactive agent from scratch. However, training on raw interaction logs (Zhu et al., 2023) is suboptimal: raw sequences reflect passive user drift rather than goal-oriented planning, leading to *goal misalignment*. To bridge this gap, we propose an trajectory mining strategy. Instead of indiscriminately slicing sequences, we distill high-quality, physically coherent *expert demonstrations* from historical logs, governed by a rigorous **Feasibility Oracle**.

---

[2]https://grouplens.org/datasets/movielens/1m/
[3]https://cseweb.ucsd.edu//~jmcauley/datasets.html#steam_data/
[4]https://cseweb.ucsd.edu/~jmcauley/datasets/amazon_v2/

---

**Algorithm 1** Goal-oriented Trajectory Mining

---

1: **Input:** User sequence $S_u$, Oracle $\mathcal{F}$, History Length $n$
2: **Output:** Expert Demonstration Set $\mathcal{D}_u$
3: Initialize current path $\tau \leftarrow \{S_u[n]\}$
4: Initialize $\mathcal{D}_u \leftarrow \emptyset$
5: **for** index $k = n + 1$ **to** $|S_u|$ **do**
6:     Let $prev = S_u[k-1]$, $curr = S_u[k]$
7:     **if** $\mathcal{F}(prev, curr) == 0$ **then**
8:        # // Lack coherence: archive the path
9:        **if** $|\tau| > 1$ **then**
10:           Let $g = \tau[-1]$ # The last item as the target
11:           $\mathcal{D}_u \leftarrow \mathcal{D}_u \cup \{(\tau, g)\}$
12:        **end if**
13:        $\tau \leftarrow \emptyset$ # Reset path
14:     **end if**
15:     Append $curr$ to $\tau$
16: **end for**
17: **Return** $\mathcal{D}_u$

---

### B.3.1. THE FEASIBILITY ORACLE

The core of our mining strategy is to ensure that every step in the training data represents a valid, smooth transition that a user would naturally accept.

**Definition B.1** (Feasibility Oracle). Let $\mathcal{I}$ be the item set. We define the Feasibility Oracle as an indicator function $\mathcal{F} : \mathcal{I} \times \mathcal{I} \rightarrow \{0, 1\}$, which evaluates whether the transition $(i_t \rightarrow i_{t+1})$ satisfies semantic coherence constraints.

To ensure generalizability, we instantiate $\mathcal{F}$ for both structured and unstructured scenarios:

**Instantiation I: Structure-based (via KG).** If a Knowledge Graph (KG) $\mathcal{G}$ is available, feasibility is grounded in explicit attribute sharing. Let $\mathcal{N}(i)$ be the set of one-hop neighbors of item $i$ in $\mathcal{G}$. A transition is feasible if:

$$\mathcal{F}_{\text{KG}}(i_t, i_{t+1}) = \mathbb{I}(|\mathcal{N}(i_t) \cap \mathcal{N}(i_{t+1})| \geq 1). \tag{31}$$

**Instantiation II: Semantics-based (via LLM).** For domains lacking structured metadata, we leverage LLMs as a proxy for human judgment on transition naturalness. We construct a verification prompt $\mathcal{P}(i_t, i_{t+1})$ and define:

$$\mathcal{F}_{\text{LLM}}(i_t, i_{t+1}) = \mathbb{I}(\text{LLM}(\mathcal{P}(i_t, i_{t+1})) = \texttt{"Yes"}). \tag{32}$$

### B.3.2. TRAJECTORY DISTILLATION PROCESS

Guided by $\mathcal{F}$, we refine history logs into a set of *expert demonstrations*. The procedure is detailed in Algorithm 1.

By strictly enforcing $\mathcal{F}$, the dataset $\mathcal{D}_{\text{expert}}$ eliminates abrupt transitions while preserving the authentic, multi-step reasoning chains found in real user behavior. This provides the model with a rich set of feasible plans to learn from before optimizing for efficiency in later stages.

### B.4. Semantic Tokenization

This subsection details the semantic ID generation process introduced in Section B.1.

### B.4.1. ITEM PROFILE GENERATION

For each item, we first build a rich item profile. Instead of relying only on raw fields (e.g., title, short description, sparse attributes), we prompt a large language model to generate a structured, high-level description of the item (e.g., key functions, typical usage scenarios, target users, style, and complementary items). This step normalizes noisy metadata and incorporates

external world knowledge, allowing items with similar semantics to be described in a consistent manner, even when their original texts are heterogeneous or incomplete. Specifically, we utilize GPT-4 as a foundation model to generate the item profile. The prompt details and item profile generated are shown in Figure 5.

---

**Profile Generation for MovieLens-1M dataset**

**Prompt:**

Introduce the movie **Titanic (1997)** and describe its attributes, including genre, director/actors, country, character, plot/theme, mood/tone, critical acclaim/award, production quality, and soundtrack.

- - - - - - - - - - - - - - - - - - - - - - - - - - - - - - - - - - - - - - - - - - - - - - - - -

**Answer:**

Titanic is a romantic disaster film released in **1997**, directed by **James Cameron**. The movie is set in 1912. The movie's plot and theme revolve around **love, sacrifice, and tragedy**.
The movie is a combination of different genres, including **romance, drama, and disaster**.
The director, James Cameron, brilliantly captures the mood and tone of the movie, portraying the luxurious and glamorous lifestyle on the Titanic ······
Overall, Titanic is a timeless classic that continues to captivate audiences with its **compelling story**, breathtaking visuals, and unforgettable soundtrack.

---

*Figure 5.* Prompt and item profile for MovieLens-1M dataset.

### B.4.2. SEMANTIC ID GENERATION

To get the semantic ID of each item, we follow the manner of existing works. We first feed the item profile into a text (or multimodal) embedding model to obtain a dense representation in a shared semantic space. Here, we use the state-of-the-art embedding model qwen3-embedding-8B (Zhang et al., 2025b) as our backbone to map the text profile into embeddings. The resulting vector captures both surface-level semantics (brands, categories, attributes) and higher-level concepts (use cases, aesthetics, user intent), which is essential for semantic retrieval and clustering.

Finally, we train a Residual Quantized VAE (RQ-VAE) on these embeddings to map each continuous vector to a compact sequence of codebook indices, i.e., a semantic ID. RQ-VAE progressively quantizes the residuals of the embedding, allowing us to represent items with a short discrete code while preserving fine-grained semantic similarity. The RQ-VAE model comprises three components: a DNN encoder that encodes the input semantic embedding into a latent representation, a residual quantizer that outputs a quantized representation, and a DNN decoder that decodes the quantized representation back into the original semantic input embedding space.

Specifically, the encoder comprises five intermediate layers of sizes 2048, 1024, 512, 256, and 128, each with ReLU activation, culminating in a final latent representation dimension of 128. To quantize this representation, five levels of residual quantization are used. For each level, a codebook of cardinality 128 is maintained, where each vector in the codebook has a dimension of 768 following the output of the qwen3-embedding-8B model (Zhang et al., 2025b). When computing the total loss, we use $\beta = 0.25$. The RQ-VAE model is trained for 10k epochs. We use Adagrad optimizer with a learning rate of 0.001 and a batch size of 2048. Upon training, we use the learned encoder and the quantization component to generate a 3-tuple Semantic ID for each item. To avoid multiple items being mapped to the same Semantic ID, we add a unique 4th code for items that share the same first three codewords, i.e. two items associated with a tuple (64, 8, 29) are assigned (64, 8, 29, 0) and (64, 8, 29, 1) respectively (if there are no collisions, we still assign 0 as the fourth codeword). This results in a unique Semantic ID of length $K = 4$ for each item in the recommendation corpus.

## C. Evaluation Metrics

This section provides detailed definitions of the evaluation metrics used in Section 4. For each subsequence in the testing set, we randomly sampled an item that the user did not interact with as the target item, which follows the setting of the existing works (Bi et al., 2024; Wang et al., 2025a;b). Finally, we report the results on the test set. We detail the four evaluation

metrics used in our experiments: IoI (Increase of Interest), IoR (Increase of Rank), CTR (i.e., HitRate), and Coherence. CTR is calculated by treating each intermediate item in the guiding sequence as a separate single-choice task. The final CTR reported is the average of these CTR across all correct answers in the sequence. IoI and IoR quantify the sequence-level user interests shift towards the target item. Coherence calculates the correlation of any adjacent items in the guiding sequence. The SASRec evaluator used for computing IoI, IoR, and CTR is trained exclusively on the complete interaction histories of training-split users. Below are the definitions of each metric, along with an example calculation.

**IoI (Increase of Interest).** IoI quantifies how much the modelled interest in the target item $i_T$ changes when an influence path $L_u$ is appended to the original interaction history $S_u$. Let $P(i \mid s)$ denote the evaluator's predicted acceptance probability of item $i$ conditioned on sequence $s$, and let $\oplus$ denote sequence concatenation. The formula of IoI is:

$$\text{IoI}(S_u, L_u, i_T) = \log P(i_T \mid S_u \oplus L_u) - \log P(i_T \mid S_u). \tag{33}$$

A positive IoI indicates that, according to the evaluator, the influence path $L_u$ increases the user's preference for the target item $i_T$ relative to using the history $S_u$ alone. Here, we pretrained the SASRec as an evaluator to calculate the $P(i \mid s)$.

**IoR (Increase of Rank).** IoR measures how much the ranking position of the target item $i_T$ improves when the influence path $L_u$ is appended to the original history $S_u$. Let $R(i \mid s)$ denote the rank of item $i$ (with 1 being the best rank) under the evaluator conditioned on sequence $s$. The IoR is defined as:

$$\text{IoR}(S_u, L_u, i_T) = R(i_T \mid S_u) - R(i_T \mid S_u \oplus L_u). \tag{34}$$

By construction, a positive IoR means that the target item moves upwards in the ranked list (i.e., becomes more prominent in the recommendation list) after incorporating the influence path $L_u$ into the user sequence. Specifically, we utilize a pretrained SASRec as an evaluator to calculate the $R(i \mid s)$.

**Coherence.** Coherence describes the correlation of the adjacent items in the sequence. We define the metric to quantify how semantically consistent a given guiding sequence $L_u = [i_1, i_2, \ldots, i_{|L_u|}]$ is. Let $\text{corr}(i, j)$ denote the correlation between two items $i$ and $j$. In our setting, $\text{corr}(i, j)$ is defined based on shared item features as shown in Formula 35.

$$\text{corr}(i, j) = \begin{cases} 1, & \text{if } i \text{ and } j \text{ share at least one common feature}, \\ 0, & \text{otherwise}. \end{cases} \tag{35}$$

The coherence of the guiding sequence $L_u$ is then defined as the average correlation over all adjacent item pairs in $L_u$:

$$\text{Coherence}(L_u) = \frac{1}{|L_u| - 1} \sum_{k=1}^{|L_u|-1} \text{corr}(i_k, i_{k+1}). \tag{36}$$

**CTR.** CTR quantifies the average interaction probability of a user $u$ with the items in a guiding sequence $L_u = [i_1, i_2, \ldots, i_{|L_u|}]$. Let $f_{\text{SASRec}}(\cdot)$ denote the SASRec (Kang & McAuley, 2018) encoder that maps an item sequence $S$ to a user embedding in $\mathbb{R}^d$. For each position $k \in \{1, \ldots, |L_u|\}$, we first construct the prefix-augmented sequence in Formula 37, where $\oplus$ denotes sequence concatenation.

$$S_u^{(k)} = \begin{cases} S_u, & k = 1, \\ S_u \oplus [i_1, i_2, \ldots, i_{k-1}], & k \geq 2, \end{cases} \tag{37}$$

The corresponding user embedding is then given by $\mathbf{h}_u^{(k)} = f_{\text{SASRec}}(S_u^{(k)}) \in \mathbb{R}^d$. Let $\mathbf{v}_i \in \mathbb{R}^d$ denote the embedding of item $i$. The predicted interaction probability between user $u$ and the $k$-th item $i_k$ in $L_u$ is computed from the inner product between $\mathbf{h}_u^{(k)}$ and $\mathbf{v}_{i_k}$ (Bi et al., 2024; Lian et al., 2025):

$$p_{u,k} = \sigma\left(\left(\mathbf{h}_u^{(k)}\right)^\top \mathbf{v}_{i_k}\right), \qquad \sigma(x) = \frac{1}{1 + e^{-x}}. \tag{38}$$

The sequence-level CTR of user $u$ on the guiding sequence $L_u$ is defined as the average interaction probability:

$$\text{CTR}(S_u, L_u) = \frac{1}{|L_u|} \sum_{k=1}^{|L_u|} p_{u,k}. \tag{39}$$

## D. Baselines

To thoroughly evaluate our proposed methods for proactive path reasoning, we conduct comprehensive evaluations across various types of methods, including both Sequential recommendation and proactive recommendation methods. The baselines are as follows:

### D.1. Sequential Recommendation Methods

- **GRU4Rec (Hidasi et al., 2015)**, a representative baseline that utilizes the standard GRU architecture to effectively encode user interaction sequences.

- **BERT4Rec (Sun et al., 2019)**, a widely-adopted model that uses bidirectional self-attention layers to incorporate deeper contextual information across user behavior sequences.

- **LightSANs (Fan et al., 2021)**, a novel approach that leverages a low-rank decomposition self-attention mechanism to efficiently capture user-item interactions.

- **FEARec (Du et al., 2023)**, a contrastive learning-based model that uses time domain attention and auto-correlation.

### D.2. Proactive Recommendation Methods

- **IRN (Zhu et al., 2023)**, a proactive recommendation paradigm that generates influence paths via a Transformer-based Influential Recommender Network with a personalized impressionability mask that controls how strongly each user is nudged.

- **IPG (Bi et al., 2024)**, an iterative preference guidance framework for proactive recommendation that items in guiding sequences are re-ranked using an explicit IPG score that jointly considers interaction probability and guiding value.

- **ITMPRec (Lian et al., 2025)**, an intention-based targeted multi-round proactive recommendation framework which iteratively nudges users toward the pre-match target item via intention-induced scoring and user-specific arousal coefficients.

- **LLM-IPP (Wang et al., 2025a)**, a LLM-based method that formulates influence path planning as a prompt-based reasoning task for large language models, enabling them to generate coherent multi-step recommendation paths that guide users from their historical interactions to a designated target item while taking into account item semantics, user intent, and transition coherence.

- **T-PRA (Wang et al., 2025b)**, a tunable LLM-based proactive recommendation agent that formulates proactive recommendation as a sequential decision-making and path-planning problem, where an LLM-based Actor–Advisor framework (Kahneman, 2011) adapts recommendations in real time based on simulated user feedback, and an LLM-based Critic with multi-objective rewards is used to perform DPO-style (Rafailov et al., 2023) agent tuning so that the learned policy optimizes long-term influence toward target items rather than only short-term accuracy.

## E. Implementation Details

This section provides implementation details for all methods evaluated in Section 4.

### E.1. Sequential Recommendation Methods

We implement BERT4Rec, GRU4Rec, CORE, LightSANs, and FEARec using the open-source recommendation library RecBole (Zhao et al., 2021). Since sequential recommendation methods are not designed for proactive tasks, to make a justified comparison, we follow the setting from (Zhao et al., 2021). The details are as follows:

Given a user's interaction history $S_u$ and a predefined objective item $i_T$, we first employ a standard sequential recommender (e.g., GRU4Rec) to generate a top-k list of next-item candidates at each step. These candidates are then re-ranked according to their distance to the objective item in the item embedding space, and the closest item is greedily appended to the influence path. The procedure is repeated until the objective item is reached or a maximum path length is exceeded. In this way, the baseline still learns user preferences and sequential dependencies in the usual next-item fashion, but the greedy re-ranking at inference time makes the resulting recommendation sequence proactively drift toward the target item.

For these models above, we ensured that key settings, such as batch size, the number of encoder blocks, and attention heads, were aligned with our model for a fair comparison. However, for other settings, we followed the recommended configurations in the original papers.

To make a fair comparison and avoid label leakage, we collect the interaction data from the user in the training set, as described in Section B.3, to train the sequential recommendation methods for validation.

### E.2. Proactive Recommendation Methods

- **IRN (Zhu et al., 2023)**: we set the mask weights to $w_t = 1$ and $w_h = 0.05$. We segment users' interaction sequences into subsequences whose length lies between $l_{\min} = 20$ and $l_{\max} = 60$. The model is trained for 200 epochs, and during inference, we generate influence paths of length 10, consistent with the original inference setup.

- **IPG (Bi et al., 2024)**: we set the preference-evolution coefficient $\gamma = 0.8$ following the original work. For user and item representation, we adopt a pretrained BERT4Rec as the backbone to compute their embeddings with 64 dimension. We iteratively generate 10 intermediate items as influence paths, consistent with the original inference setup.

- **ITMPRec (Lian et al., 2025)**: We adopt the item embeddings learned by BERT4Rec and perform k-means clustering with $k = 256$ to obtain the intention vectors $C$. The personalized preference-evolution coefficient $\gamma_u$ is thresholded with a cutoff of 0.2. The intention-level coefficient is set to $\lambda = 0.1$, and for the top-$n$ pre-selection stage before re-ranking, we follow the original work and choose the top 10 items. We generate influence paths comprising 10 items which follows the original inference setup.

- **LLM-IPP (Wang et al., 2025a)**: we adopt Llama-3.1-8B-Instruct (Dubey et al., 2024) as the backbone model to generate the guiding sequences, and use the Tree-of-Thought prompt variant, which was reported to achieve the best performance in the original work.

- **T-PRA (Wang et al., 2025b)**: we follow the original hyperparameter settings: we use Llama-3.1-8B-Instruct (Dubey et al., 2024) as the base model for all agents, and fine-tune them with LoRA (Hu et al., 2022) of rank 8 applied to all transformer modules. The models are trained for 5 epochs per dataset with a learning rate of $5 \times 10^{-5}$, a cosine learning-rate scheduler with a warm-up ratio of $10\%$, and 8 gradient accumulation steps. During inference, we set the decoding temperature to 0.5.

### E.3. ProRL

ProRL undergoes a two-stage framework including pretraining and reinforcement learning. To obtain a strong prior $\pi_0$, we pretrain a T5-based sequence-to-sequence model on carefully constructed Smooth-Guided paths elaborated in Section B.3. As described in Section B.1, the model operates on semantic ID sequences: each item is represented by $K = 4$ tokens, and the policy autoregressively generates these tokens. This pretraining stage provides: a semantic prior that constrains the action space, planning capability for guiding the path, and a foundation for efficient RL fine-tuning.

In the RL stage, we explore the optimal strategy leading to both path feasibility and guiding effectiveness by SRC and PSAE. The rewards are computed at the item level after decoding the generated token sequences back to items, and the gradients are distributed to all tokens within each item according to Eq. (30).

**Pretraining.** We implement a lightweight encoder-decoder Transformer backbone adapted from T5 (Raffel et al., 2020). The model is configured with a shallow depth, utilizing three layers for both the encoder and the decoder. Regarding the attention mechanism, we employ four heads with a head dimension of 64. We set the hidden dimension to 128 and the intermediate feed-forward network dimension to 512, while using ReLU for activation. This configuration yields a highly efficient model with approximately 1.97M parameters (excluding embeddings).

*Table 10.* Hyperparameter settings categorized by training stages.

| Stage | Hyperparameter | MovieLens-1M | Steam | Amazon-Book |
|---|---|---|---|---|
| **Common** | num_layers | 3 | 3 | 3 |
| | d_model | 128 | 128 | 128 |
| | d_ff | 512 | 512 | 512 |
| | num_heads | 4 | 4 | 4 |
| | d_kv | 64 | 64 | 64 |
| | dropout_rate | 0.1 | 0.1 | 0.1 |
| | optimizer | adamw | adamw | adamw |
| **Pretrain** | learning_rate | 0.005 | 0.005 | 0.005 |
| | batch_size | 1,024 | 1,024 | 1,024 |
| | max_epochs | 200 | 200 | 200 |
| | warmup_steps | 10,000 | 10,000 | 10,000 |
| | vocabulary_size | 1,026 | 1,026 | 1,026 |
| **RL** | learning_rate | 1e-4 | 1e-5 | 5e-4 |
| | batch_size | 128 | 128 | 128 |
| | num_return_samples | 16 | 16 | 16 |
| | temperature | 1 | 1 | 1 |
| | kl_coeff | 0.01 | 0.01 | 0.01 |
| | RL_epochs | 50 | 50 | 50 |
| | $\alpha$ | 1 | 1 | 1 |
| | $\beta$ | 1 | 1 | 1 |
| | $\gamma$ | 1 | 1 | 1 |

*Table 11.* Experimental Results on Different Datasets

| Dataset | Model | CTR | Coherence | IoI | IoR |
|---|---|---|---|---|---|
| MovieLens-1M | w SmGD | 0.8671 | **0.7488** | **0.8600** | **254.42** |
| | w/o SmGD | **0.9110** | 0.5522 | -0.0531 | 75.23 |
| Steam | w SmGD | 0.7453 | **0.9493** | **0.4230** | **101.15** |
| | w/o SmGD | **0.7787** | 0.8051 | 0.2598 | 64.63 |
| Amazon-Book | w SmGD | **0.6410** | **0.6885** | **0.1650** | **72.91** |
| | w/o SmGD | 0.6155 | 0.5506 | 0.1026 | 17.01 |

**Reinforcement Learning.** Following the pretraining phase, we employ reinforcement learning to further refine the policy $\pi$, shifting the model's focus from mere sequence imitation to goal-oriented trajectory optimization. The reward signal comprises two primary components: a feasible reward that ensures user acceptance of intermediate items, and a guidance reward that quantifies the effectiveness of guidance.

To address the high variance and sparse signals inherent in long-sequence generation, we employ Stepwise Reward Centering. This technique dynamically adjusts the baseline for rewards at each step, effectively eliminating the length bias incurred by length manipulation. Furthermore, we implement Position-Specific Advantage Estimation, which assigns credit more precisely by considering the temporal importance of each item in the guidance path. During the fine-tuning process, we optimize the policy using a policy gradient objective, incorporating a KL-divergence constraint relative to $\pi_0$ to prevent the model from collapsing into suboptimal, repetitive paths.

This two-stage approach enables ProRL to generate trajectories that are not only highly reachable for users but also strategically aligned with the intended guidance objectives.

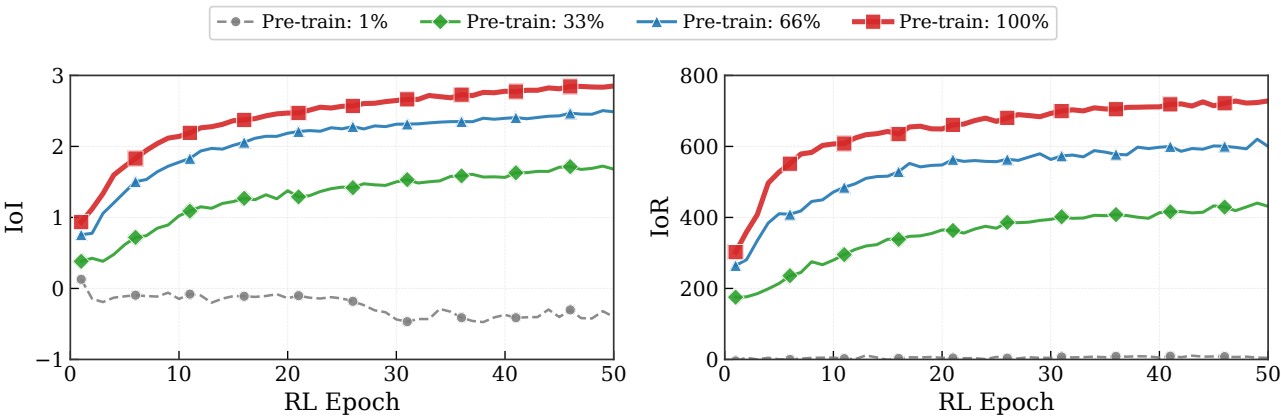

*Figure 6.* The impact of pre-training maturity on reinforcement learning efficiency. We evaluate the performance of ProRL across different stages of the pre-training process (1%, 33%, 66%, and 100% completion) on the MovieLens-1M dataset. The results indicate that a sufficiently converged semantic prior is a prerequisite for effective RL optimization, as it effectively constrains the action space and mitigates the sparse reward challenge in proactive guidance.

**A2C Baseline Implementation.** For the A2C baseline in Section 4.3.3, we implement a critic network as a 2-layer MLP with 256 hidden units. Since a randomly initialized critic invariably causes training collapse, we warm up the critic for 5 epochs before joint actor-critic training. We search over the critic loss coefficient in $\{0.1, 0.25, 0.5, 1.0\}$, with all other settings identical to ProRL. We report the best results (coefficient 0.25).

**Uniform Sampling Budget.** In the gradient estimator comparison (Section 4.3.3), all five methods (RF, RTG, GRPO, A2C, and ProRL) sample the same $m=16$ rollout trajectories per input, corresponding to num_return_samples in Table 10. Although methods such as REINFORCE do not inherently require multiple rollouts to form their gradient estimates, we uniformly draw 16 trajectories for every method and average over them when computing the policy gradient (cf. Eq. (3), (7), and (9)). This ensures that all estimators operate under an identical computational budget, so the observed performance differences are attributable to the estimator design itself rather than to differences in sampling cost.

## F. Supplementary Experiments

This section presents additional experiments including ablation studies, sensitivity analyses, and cross-environment evaluations to further validate the ProRL framework.

### F.1. Smooth-Guided Data Construction

To validate the need to utilize the Smooth-Guided Data (SmGD) described in Section B.3 for pretraining, we conducted an ablation study comparing our approach with the data processing method proposed in (Zhu et al., 2023). The results are shown in Table 11. Comparative results demonstrate that the model pretrained on SmGD outperforms the model trained on randomly sliced data across most proactive recommendation metrics. This confirms that semantically coherent training data is crucial for effective guidance.

### F.2. Impact of Pre-training Initialization

To investigate the dependence of reinforcement learning on the quality of semantic priors, we analyzed the performance of ProRL when initialized with checkpoints from different stages of the pre-training process (1%, 33%, 66%, and 100%). As illustrated in Figure 6, the agent initialized with only minimal pre-training fails to learn a meaningful policy, confirming that the sparsity of successful guidance signals in the high-dimensional action space renders cold-start exploration infeasible. Conversely, we observe a strict positive correlation between the maturity of the supervised prior and the efficiency of the RL phase. This demonstrates that robust supervised pre-training is not merely a warm-up but a foundational prerequisite, constructing a semantic map that narrows the action space and enables the agent to effectively optimize for long-term

strategic guidance.

## F.3. Model Robustness Analysis

Target selection in proactive recommendation generally prioritizes either random items (Zhu et al., 2023; Bi et al., 2024; Wang et al., 2025a;b) or those with high user interaction potential (Lian et al., 2025). To verify the robustness of our approach, we implement two selection schemes: Random Selection and Filtered Selection. In the former, we randomly assign a non-interacted item as the target. In the latter, we score candidate items based on predicted interaction willingness and select those ranked at the 20th, 40th, and 60th percentiles as targets. This design enables us to evaluate the guidance capabilities of our method against baselines under varying degrees of target difficulty.

Specifically, we compare against FEARec (best sequential model in Table 1) and proactive methods (IPG, ITMPRec, LLM-IPP) across all three datasets. The results in Figure 7 consistently demonstrate that ProRL achieves superior robustness.

### F.3.1. PERFORMANCE SUPERIORITY ACROSS DIVERSE METRICS

Robustness in proactive recommendation requires a model to maintain high user satisfaction while effectively executing guidance goals. ProRL demonstrates a "Pareto dominance" over existing methods across all four key dimensions:

- **User Engagement Preservation (CTR & Coherence):** Unlike baseline models that often sacrifice user experience to force guided items, ProRL maintains exceptionally high engagement metrics. On the dense *MovieLens-1M* dataset, our model sustains a Click-Through Rate (CTR) of approximately $0.89$ and a Semantic Coherence of $0.95$ across all intervention ratios. In contrast, strong baselines like FEARec only achieve CTRs in the range of $0.55$ to $0.60$. Even on the sparse *Steam* dataset, where maintaining coherence is challenging, ProRL achieves a Coherence score of $> 0.8$, significantly outperforming purely generative baselines, such as ITMPRec ($0.65$ on average). This indicates that the latent space editing mechanism of ProRL successfully preserves the user's inherent preference manifold while injecting guided items.

- **Guidance Efficacy and Impact (IoI & IoR):** On the *Steam* dataset, most baselines exhibit negative IoI values, indicating that guided items disrupt natural item transitions. ProRL consistently maintains positive IoI scores. In terms of IoR, ProRL achieves scores between $1300$ and $1500$ on *Steam*, an order of magnitude higher than standard baselines (typically $< 200$).

### F.3.2. STABILITY UNDER VARYING INTERVENTION INTENSITIES

A robust proactive recommender must remain stable regardless of the aggressiveness of the guidance signal. We analyzed the performance variance under different target ratios ($20\%$, $40\%$, $60\%$) and a stochastic setting:

- **Insensitivity to Guidance Pressure:** Standard proactive models often suffer from performance degradation as the guidance target ratio increases (e.g., forcing $60\%$ of items to be from a target set). However, ProRL exhibits remarkable stability. On the *MovieLens-1M* dataset, as the ratio increases from $20\%$ to $60\%$, the fluctuation in Coherence is minimal (staying above $0.94$), whereas competitive baselines like ITMPRec see a sharper decline. This suggests that ProRL's gradient-based perturbation finds optimal injection points that are resilient to the quantity of guided items.

- **Resilience to Random Targets:** The *Random* setting serves as a stress test with unpredictable guidance goals. ProRL adapts seamlessly, achieving a CTR of $0.547$ and Coherence of $0.89$ on *MovieLens-1M*, matching or exceeding fixed-ratio scenarios. This confirms that ProRL learns a robust policy rather than overfitting to a specific intervention pattern.

### F.3.3. ADAPTABILITY TO DATA CHARACTERISTICS

ProRL's consistent top performance across domains with varying data densities, from the sparse *Steam* to the dense *MovieLens-1M*, confirms that its core mechanism is domain-agnostic.

## F.4. Performance on Unseen evaluators: Full Results

To show the generalization ability of our methods, we evaluate the performance on GRU4Rec, BERT4Rec and LightSANs as unseen evaluators during training process. The results of the BERT4Rec and LightSANs are shown in Table 12, Table 13

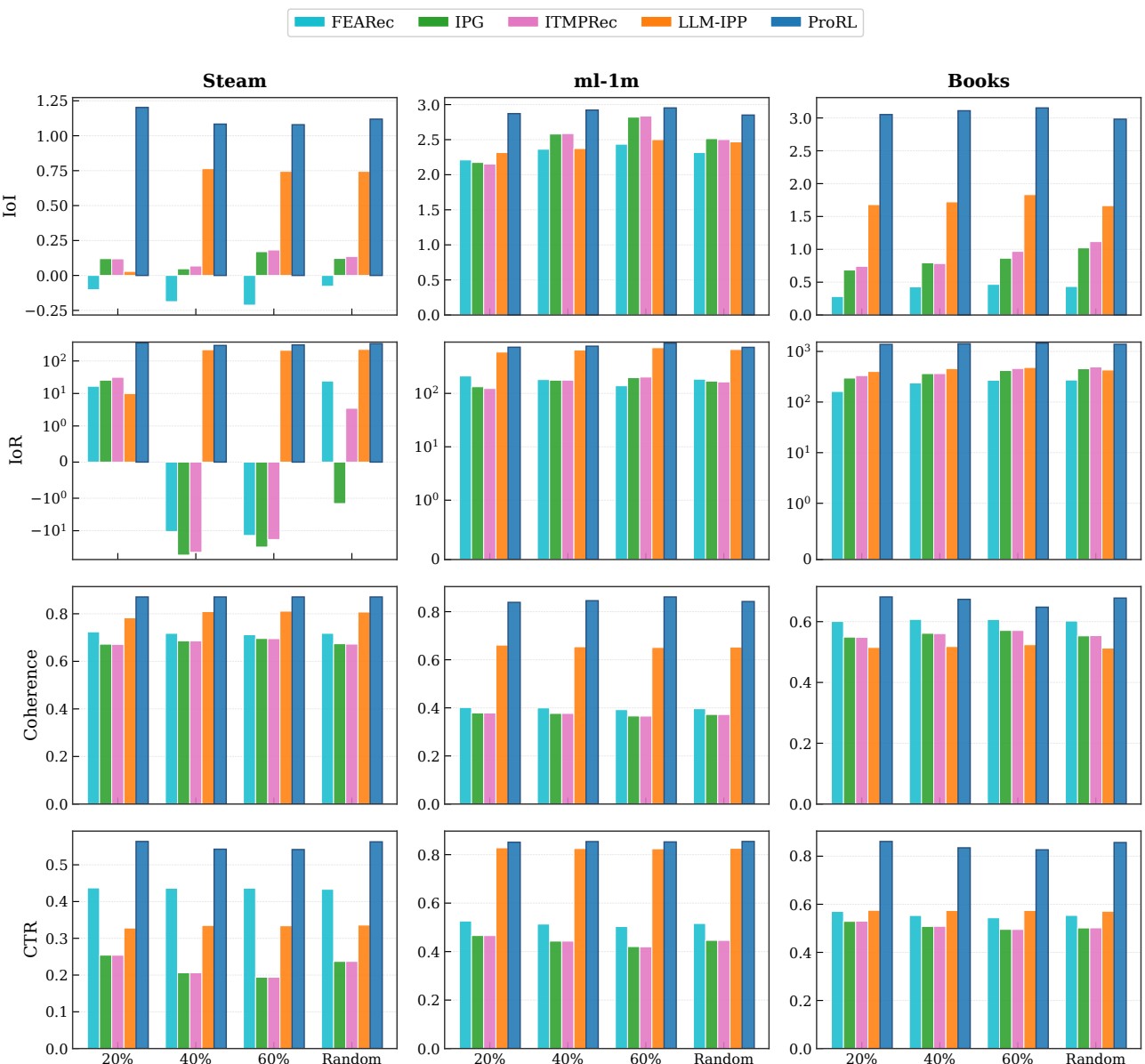

*Figure 7.* Robustness analysis across varying target selection schemes and guidance difficulties. We evaluate the performance on three datasets under Random Selection and Filtered Selection (20th, 40th, and 60th percentiles of interaction willingness). Higher percentiles represent higher guidance difficulty (lower user interest). Our method consistently outperforms baselines across all metrics, CTR, Coherence, IoI, and IoR, demonstrating superior robustness regardless of target accessibility.

respectively.

### F.5. Alternative Approaches to Eliminating the Length Shortcut

Section 3.2 introduces Stepwise Reward Centering, which eliminates the length shortcut by subtracting the empirically estimated expected step reward. A natural question arises: can we achieve the same effect through manual hyperparameter tuning instead of data-driven estimation?

**Alternative Approach: Fixed Offset.** We consider a simplified alternative where a fixed offset $\epsilon$ is subtracted from the

*Table 12.* Proactive Recommendation performance of all models on different datasets (BERT4Rec as evaluator) in terms of CTR (i.e., HitRate), Coherence, IoI, and IoR. The best performances are highlighted in bold. The superscript * indicates the Improvement is statistically significant, where the p-value is less than 0.05.

| Dataset | MovieLens-1M | | | | Steam | | | | Amazon-Book | | | |
|---|---|---|---|---|---|---|---|---|---|---|---|---|
| Model | CTR | Coherence | IoI | IoR | CTR | Coherence | IoI | IoR | CTR | Coherence | IoI | IoR |
| GRU4Rec | 0.5914 | 0.3717 | 2.0435 | 69.08 | 0.4716 | 0.7026 | -0.0863 | -8.44 | 0.5748 | 0.5838 | -0.0554 | 100.99 |
| LightSANs | 0.5995 | 0.3957 | 2.0493 | 83.29 | 0.4556 | 0.7150 | -0.0784 | -13.41 | 0.5783 | 0.5934 | 0.0865 | 165.33 |
| FEARec | 0.5849 | 0.3964 | 2.1734 | 109.23 | 0.4509 | 0.7177 | -0.0937 | -11.39 | 0.5637 | 0.6020 | 0.1803 | 231.99 |
| IRN | 0.7688 | 0.4706 | 2.2364 | 121.64 | 0.3740 | 0.6698 | -0.5034 | -10.15 | 0.5607 | 0.5477 | 0.0217 | 111.62 |
| IPG | 0.4887 | 0.3725 | 2.5595 | 146.41 | 0.2246 | 0.6740 | 0.1017 | 11.43 | 0.5072 | 0.5531 | 1.0802 | 548.84 |
| ITMPRec | 0.4821 | 0.3714 | 2.5632 | 150.00 | 0.2262 | 0.6725 | 0.1117 | 11.73 | 0.5068 | 0.5540 | 1.0939 | 552.92 |
| LLM-IPP | 0.6540 | 0.6288 | 2.2720 | 85.45 | 0.3424 | 0.8022 | -0.4542 | -12.19 | 0.5709 | 0.5132 | 0.2681 | 176.14 |
| T-PRA | 0.4612 | 0.3415 | 2.4502 | 220.75 | 0.3012 | 0.7399 | 0.2215 | 24.12 | 0.5024 | 0.4418 | 0.6588 | 323.12 |
| ProRL (Ours) | **0.8403**\* | **0.8422**\* | **2.6111** | **699.03**\* | **0.4805**\* | **0.8707**\* | **0.4258**\* | **68.27**\* | **0.8192**\* | **0.6823**\* | **2.7400**\* | **1290.14**\* |

*Table 13.* Proactive Recommendation performance of all models on different datasets (LightSANs as evaluator) in terms of CTR (i.e., HitRate), Coherence, IoI, and IoR. The best performances are highlighted in bold. The superscript * indicates the Improvement is statistically significant, where the p-value is less than 0.05.

| Dataset | MovieLens-1M | | | | Steam | | | | Amazon-Book | | | |
|---|---|---|---|---|---|---|---|---|---|---|---|---|
| Model | CTR | Coherence | IoI | IoR | CTR | Coherence | IoI | IoR | CTR | Coherence | IoI | IoR |
| GRU4Rec | 0.4136 | 0.3717 | 1.5489 | 102.42 | 0.4275 | 0.7026 | 0.0391 | 27.93 | 0.5527 | 0.5838 | 0.1417 | 119.23 |
| BERT4Rec | 0.4432 | 0.3889 | 1.2425 | 73.62 | 0.4444 | 0.7390 | 0.0927 | 26.23 | 0.5662 | 0.5591 | 0.1623 | 113.93 |
| FEARec | 0.4126 | 0.3964 | 1.7654 | 153.61 | 0.4189 | 0.7177 | -0.0104 | 9.07 | 0.5489 | 0.6020 | 0.5008 | 227.05 |
| IRN | 0.6812 | 0.4706 | 1.9027 | 188.26 | 0.3452 | 0.6698 | 0.0240 | 8.87 | 0.5334 | 0.5477 | 0.1913 | 131.60 |
| IPG | 0.3417 | 0.3725 | 2.1786 | 182.28 | 0.2354 | 0.6740 | 0.1361 | 35.56 | 0.5401 | 0.5531 | 0.5428 | 255.12 |
| ITMPRec | 0.3323 | 0.3714 | 2.2083 | 187.68 | 0.2325 | 0.6725 | 0.1440 | 39.88 | 0.5427 | 0.5540 | 0.5585 | 239.98 |
| LLM-IPP | 0.7722 | 0.6288 | 2.5571 | 681.90 | 0.3198 | 0.8022 | 0.9927 | 237.36 | 0.5651 | 0.5132 | 1.5765 | 430.93 |
| T-PRA | 0.5128 | 0.3415 | 2.6012 | 712.21 | 0.2617 | 0.7399 | 1.0823 | 220.12 | 0.5012 | 0.4418 | 1.7812 | 502.98 |
| ProRL (Ours) | **0.8090**\* | **0.8422**\* | **2.9820**\* | **755.83**\* | **0.5239**\* | **0.8707**\* | **1.3722**\* | **306.12**\* | **0.8912**\* | **0.6775**\* | **2.8851**\* | **1286.74**\* |

variance-normalized step reward:

$$\tilde{r}_t = \sum_{i=1}^{K} w_i \cdot \frac{r_t^{(i)}}{\sigma^{(i)}} - \epsilon, \tag{40}$$

where $\sigma^{(i)}$ is the standard deviation of the $i$-th reward component. Unlike Eq. (6), this formulation omits the mean subtraction $\mu^{(i)}$ and instead relies on a manually determined offset $\epsilon$ to neutralize the positive bias in step rewards. By tuning $\epsilon$, one might hope to manually achieve zero expected gain from path extension.

**Experimental Setup.** In multi-objective settings, the interaction between multiple reward components would make offset tuning even more complex and unstable. To give this alternative approach its best chance, we simplify the evaluation by using IoI as the sole reward signal on the Amazon-Book dataset. All other training hyperparameters remain identical to the main experiments. We vary the offset $\epsilon \in \{0.0, -0.2, -0.4, -0.6, -0.8, -1.0\}$ and monitor the average path length of rollouts during the first 10 epochs of RL training.

**Results.** Figure 8 reveals the extreme sensitivity of this approach. When $\epsilon$ is small (close to 0, light orange curves), the positive bias in step rewards persists, and the model rapidly converges to maximum-length paths ($L \approx 10$), exhibiting the length shortcut phenomenon described in Section 2.2. As $\epsilon$ increases in magnitude, a phase transition occurs: at $\epsilon \approx -0.8$, paths collapse to minimal length ($L \approx 1$), and at $\epsilon = -1.0$, the model generates near-empty paths ($L \approx 0$). Between these extremes, intermediate values of $\epsilon$ (e.g., $-0.6$) produce unstable behavior, since path length varies significantly across epochs rather than converging to a stable value.

**Implications.** These results demonstrate that even in the simplified single-reward setting, the effective operating region for manual offset tuning is extremely narrow. A small miscalibration leads to either the original length shortcut (overlong paths) or the opposite failure mode (trivially short paths). In practice, multi-objective rewards would introduce additional complexity, making robust offset selection even more challenging. In contrast, ProRL (dark blue starred curve) achieves stable, reasonable path lengths ($L \approx 3$–$4$) without any manual tuning. By estimating $\mu^{(i)}$ from rollouts collected during

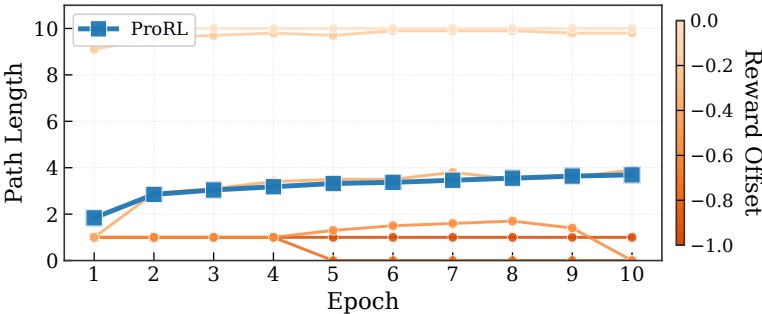

*Figure 8.* Sensitivity analysis of fixed reward offset on Amazon-Book using IoI as the sole reward. The color gradient indicates offset magnitude (darker = more negative). Small offsets (light orange) lead to maximum-length paths (length shortcut), while large offsets (dark orange) cause path collapse to near-zero length. ProRL (blue stars) achieves stable, moderate path lengths through data-driven reward centering without manual tuning.

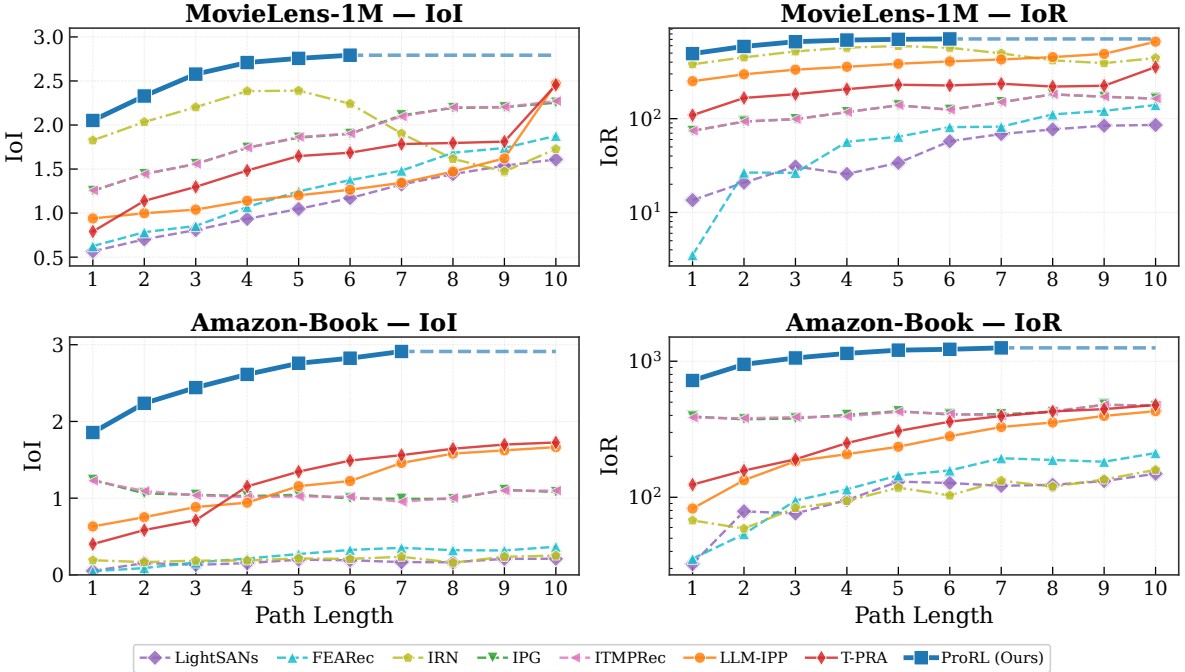

*Figure 9.* Performance comparison across varying path lengths on the MovieLens-1M (A, B) and Amazon-Book (C, D) datasets.

the first training epoch and freezing the estimates thereafter, Stepwise Reward Centering automatically calibrates to the actual reward distribution without manual tuning, ensuring that path extension yields zero expected gain throughout training. This data-driven approach eliminates the need for sensitive hyperparameter search and provides robust performance across different reward configurations and datasets.

## F.6. Decision Quality Evaluation

To evaluate the quality of local decisions, we compare performance at each path length, as shown in Figure 9. ProRL consistently outperforms baselines across all steps. Unlike baselines that rely on prolonged interactions to slowly accumulate preference shifts, ProRL ensures that every step contributes meaningfully. By addressing the length shortcut and high gradient variance, ProRL maximizes the utility of each step, demonstrating that superior path-level performance is built on effective optimization at every position.

