# OpenReview forum: "ProRL: Effective Reinforcement Learning for Proactive Recommendation via Rectified Policy Gradient Estimation"
_ICML.cc/2026/Conference — ICML 2026 regular_

### Official Review · Reviewer_rPmY · 2026-03-06

**Soundness:** 3
**Presentation:** 3
**Significance:** 2
**Originality:** 3
**Overall Recommendation:** 4
**Confidence:** 3

**Summary:**

This paper proposes ProRL, a reinforcement-learning framework for proactive recommendation, where the policy generates a path of intermediate items to gradually shift user preference toward a target item. The authors identify and analyze two policy-gradient failures—length shortcut (path extension dominates learning) and high gradient variance—and introduce Stepwise Reward Centering (SRC) and Position-Specific Advantage Estimation (PSAE) to rectify gradient estimation, showing gains on three datasets against sequential, heuristic, and LLM-based proactive baselines.

**Compliance With Llm Reviewing Policy:**

Affirmed.

**Final Justification:**

The rebuttal adequately addressed my main concerns, especially regarding evaluator robustness and path-length comparability. Since my original score was already in the accept range, the rebuttal mainly increases my confidence in that assessment rather than changing my overall judgment. Therefore, I keep my score unchanged.

**Key Questions For Authors:**

See the Strengths And Weaknesses.

**Limitations:**

See the Strengths And Weaknesses.

**Strengths And Weaknesses:**

**Soundness**

---

(1) Strengths.

The authors clearly problematize the length-dependent bias in PRS and provide a theoretical characterization of “length collapse” (stop-only model; Theorem 2.1), going beyond empirical observation.
The proposed fixes are well-motivated: SRC enforces zero expected gain from path extension, and PSAE leverages reward decomposition to use position-specific baselines, reducing variance without a learned critic (GRPO-inspired).

(2) Weaknesses

A main concern is evaluation dependence on a simulator (SASRec); while simulator-based evaluation is standard in PRS, RL can over-optimize the evaluator. The paper partially mitigates this via cross-evaluator generalization (GRU4Rec/LightSANs as unseen evaluators), which is a strong addition, but more robustness checks (more evaluators / seeds) would further strengthen the claim.




**Presentation**

---

(1) Strengths.

The narrative is easy to follow: failure case → analysis → rectified estimators → experiments/ablations.
Implementation details and hyperparameters are reasonably documented in the appendix.

(2) Weaknesses.

The multi-objective reward uses weights, but the paper mostly treats them as “default,” without clearly listing concrete values/schedules in the main discussion.


**Significance**


---


(1) Strengths.

Proactive recommendation (guiding users toward a target through intermediate items) is practically meaningful for promotion/exploration settings, and the "length shortcut" failure mode is a real bottleneck for RL-style optimization in this structured generation problem.

(2) Weaknesses.

The empirical setting remains offline and evaluator-driven, so real user impact is uncertain even with cross-evaluator tests.In addition, there is a fairness / comparability concern around path-length budgets: several baselines are run with length-20 paths following their original setups, while ProRL’s maximum-length regime appears to be around 10 (the paper repeatedly discusses "maximum-length paths (L =10)"). A fixed-budget comparison or a length–performance trade-off curve would make conclusions more robust.


**Originality**

---


(1) Strengths.

The key novelty is not “inventing REINFORCE baselines,” but a problem-structured diagnosis (length shortcut from positive-mean step rewards) plus a task-specialized rectification: SRC (centering/normalizing step rewards) and PSAE (position-specific advantage using reward-to-go and per-position group baselines). This is a meaningful, well-targeted adaptation of classic variance-reduction ideas to PRS.

(2) Weaknesses.

The authors may want to position the contribution explicitly as "problem-specific rectification and combination" rather than a fundamentally new policy gradient theory, since the components are closely related to established control-variate / baseline techniques (though applied in a novel PRS structure).

---

> ### Author Rebuttal · Authors · 2026-03-29
>
> We sincerely thank Reviewer rPmY for the thoughtful review. We appreciate the recognition of theory (Soundness: 3) and novel problem-structured diagnosis (Originality: 3). We address each concern below.
> # (Q1) Soundness: More evaluators and seeds.
> We have conducted both experiments as suggested.
>
> **1. Additional evaluators**
>
> Beyond GRU4Rec and LightSANs (Section 4.5), we add BERT4Rec as a third evaluator. Below are results on ML-1M, comparing ProRL with T-PRA (the strongest baseline):
>
> |Evaluator|Method|CTR|IoI|IoR|
> |-|-|-|-|-|
> |GRU4Rec|T-PRA|0.476|2.317|210.2|
> |-|ProRL|**0.846**|**2.456**|**649.3**|
> |LightSANs|T-PRA|0.513|2.601|712.2|
> |-|ProRL|**0.809**|**2.982**|**755.8**|
> |BERT4Rec|T-PRA|0.461|2.450|220.8|
> |-|ProRL|**0.840**|**2.611**|**699.0**|
>
> On Books and Steam, ProRL also outperforms all baselines across all four evaluators. We provide full results for all baselines on Books across evaluators in our response to Reviewer QQ1T (W2), and complete results across all baselines, datasets, and evaluators at https://anonymous.4open.science/r/Rebuttal_ProRL-F05B
>
> **2. Multi-seed stability**
>
> We train ProRL with 5 random seeds (1,2,3,4,5) and evaluate across evaluators:
>
> |Dataset|Evaluator|IoI|IoR|CTR|
> |-|-|-|-|-|
> |ML-1M|SASRec|2.838±0.020|725.2±4.6|0.855±0.004|
> |-|GRU4Rec|2.478±0.020|653.2±4.8|0.845±0.002|
> |-|LightSANs|3.001±0.025|753.1±3.6|0.811±0.003|
> |Steam|SASRec|1.102±0.021|338.6±4.1|0.565±0.003|
> |-|GRU4Rec|0.199±0.006|82.5±1.6|0.637±0.004|
> |-|LightSANs|1.335±0.022|303.3±4.6|0.524±0.003|
> |Books|SASRec|3.009±0.096|1441.4±96.0|0.850±0.009|
> |-|GRU4Rec|1.776±0.018|1015.2±36.4|0.882±0.002|
> |-|LightSANs|2.907±0.037|1276.3±24.8|0.891±0.003|
>
> Note that std/mean remains mostly $< 0.03$, confirming strong stability across seeds and evaluators.
> # (Q2) Presentation: Reward weights.
> Thanks for raising this. By Eq. 6, each reward component is normalized to zero mean and unit variance, eliminating their scale differences, so the most natural choice is equal weighting $w_i=1$ for all $i$. This is what "default weights" refers to.
>
> Our experiments validate their effectiveness: with these weights, ProRL improves all metrics rather than part of them (Table 1). We will state these values explicitly in the revision.
> # (Q3) Significance: Path-length fairness.
> We thank the reviewer for this important observation and provide both a clarification and an additional experiment.
>
> **1. Clarification**
>
> Table 1 evaluates all methods under a unified $L_{max}=10$. Appendix E.2 lists $L_{max}=20$ as each baseline's original default value, but the actual evaluation in our experiments uses $L_{max}=10$ throughout, consistent with the $L_{max}=10$ setting discussed in Section 4. We apologize for the ambiguous wording and will revise Appendix E.2 to state the protocol explicitly, and we promise to release baseline code upon acceptance for verification.
>
> **2. Additional experiment**
>
> To directly address this concern, we re-ran all methods at $L_{max}=20$. Results on MovieLens-1M (SASRec evaluator):
>
> |Method|CTR|IoI|IoR|
> |-|-|-|-|
> |IRN|0.826|2.761|568.1|
> |IPG|0.432|2.513|365.2|
> |ITMPRec|0.430|2.500|388.1|
> |LLM-IPP|0.561|2.791|692.1|
> |T-PRA|0.488|2.486|335.1|
> |**ProRL**|**0.867**|**2.885**|**735.4**|
>
> While all baselines improve with $L_{max}=20$, ProRL still exceeds every baseline. Notably, ProRL's metrics are close to results in Table 1 (with budget $L_{max}=10$), indicating it already learns to stop at optimal positions without needing extra budget. This confirms the improvement stems from superior per-step decision quality rather than path length.
>
> **3. Length–performance trade-off curve**
>
> Figure 12 (Appendix F.6) provides the suggested trade-off curve: ProRL leads at every step from 1 to 10, demonstrating that its advantage is consistent across path lengths rather than concentrated at later steps.
> # (Q4) Significance: Offline evaluation.
> We agree that offline evaluation has limitations in capturing real user behavior. This is a shared constraint of the entire proactive recommendation field: all existing PRS methods rely on offline evaluators due to the difficulty of deploying real-time interactions. Within this established framework, ProRL has been validated across 4 independent evaluators (Section 4.5 and Q1 above), 5 random seeds (Q1), and 3 datasets, representing the most comprehensive offline verification in PRS to date. We thank the reviewer for highlighting online evaluation and will discuss online A/B testing as an important future direction in the limitations.
> # (Q5) Originality: Contribution positioning.
> We appreciate this advice and agree. The core contribution of ProRL is best described as: (1) a problem-structured diagnosis identifying why standard policy gradients fail specifically in PRS (the length shortcut), and (2) a task-specialized rectification that adapts classical variance-reduction techniques to the unique reward decomposition structure of proactive recommendation. We will revise the framing in the revision.

---

> > ### Author Rebuttal · Reviewer_rPmY · 2026-04-04
> >
> > My concerns have been adequately addressed by the rebuttal. Specifically:
> > (1) Robustness (Q1): The authors provided additional experiments with BERT4Rec as a third unseen evaluator and multi-seed stability results (5 seeds across 3 datasets × 3-4 evaluators). The low std/mean ratios confirm strong reproducibility, and the consistent gains across all four evaluators represent the most comprehensive offline verification in the PRS literature to date. This fully resolves my concern about the need for more evaluators and seeds.
> >
> >
> > (2) Path-length fairness (Q3): This was my most significant concern. The authors clarified that Table 1 already uses a unified L=10 for all methods, and the ambiguity was in Appendix E.2's wording. More importantly, the new L=20 experiment shows ProRL still exceeds every baseline, while ProRL's own metrics remain close to its L=10 results. Combined with Figure 12 showing ProRL leading at every step from 1 to 10, this convincingly demonstrates that the improvement stems from per-step decision quality rather than path-length exploitation.
> >
> >
> > (3) Reward weights (Q2): The equal-weighting rationale after per-component normalization (Eq. 6) is reasonable, and the authors committed to stating this explicitly in revision.
> >
> >
> > (4) Offline evaluation (Q4): I acknowledge that offline evaluation is a shared constraint of the entire PRS field. Within this framework, the breadth of validation (4 evaluators, 5 seeds, 3 datasets) is sufficient.
> >
> >
> > (5) Contribution positioning (Q5): The authors agreed to reframe the contribution as problem-specific rectification rather than fundamentally new policy gradient theory, which aligns with my suggestion.
> >
> >
> > Overall, the rebuttal was thorough and well-supported with concrete experimental evidence. I maintain my positive assessment of the paper's strengths—the clear problem diagnosis (length shortcut), theoretically grounded rectifications (SRC + PSAE), and strong empirical results—and I keep my score accordingly.

---

> > > ### Author Response · Authors · 2026-04-04
> > >
> > > Dear Reviewer rPmY,
> > >
> > > Thank you sincerely for your thorough and positive acknowledgment. We are very grateful for your careful evaluation and for confirming that our additional experiments and clarifications adequately addressed your concerns. Your constructive feedback has genuinely helped improve this work.
> > >
> > > We noticed the system reflects option (b), which mentions follow-up questions. If there are any remaining points you would like us to clarify, please let us know so we can promptly address them. Alternatively, if you feel the discussion is satisfactorily concluded as your kind comments suggest, we would be incredibly grateful if you might consider updating the status to option (a). Thank you again for the time and effort you have devoted to reviewing our paper.
> > >
> > > Best regards,
> > >
> > > The Authors

---

### Official Review · Reviewer_QQ1T · 2026-03-12

**Soundness:** 2
**Presentation:** 2
**Significance:** 3
**Originality:** 3
**Overall Recommendation:** 4
**Confidence:** 4

**Summary:**

The authors study how to effectively apply reinforcement learning (RL) to guide user preferences via a sequence of intermediate recommendations. The work analyzes the length shortcut arising from the structural property that path-level rewards in proactive recommendation decompose into step-level increments with a positive mean, thereby coupling expected return to path length. The paper identifies two deficiencies in applying standard policy gradient methods to Proactive Recommender Systems (PRSs): (1) a length-dependent bias that causes the policy to degenerate by extending paths rather than exploring quality, and (2) high gradient variance from weighting each step by the entire path-level reward. To address these, the authors propose an RL framework with two mechanisms: Stepwise Reward Centering, which subtracts the empirically estimated per-step mean reward to enforce zero expected gain from path extension, and Position-Specific Advantage Estimation, which computes step-adapted baselines using the decomposition structure of path rewards to reduce variance. A formal theorem characterizes the rate at which stopping probability collapses under positive-mean step rewards. Experiments on three datasets (MovieLens-1M, Steam, Amazon-Book) show consistent improvements over a broad range of baselines across various metrics.

**Compliance With Llm Reviewing Policy:**

Affirmed.

**Final Justification:**

The rebuttal meaningfully addresses my two main concerns. I am raising my score, contingent on the discussed revision. See the Rebuttal Acknowledgement.

**Key Questions For Authors:**

1. Evaluation Leakage

Can the authors confirm whether the SASRec evaluator was trained only on training-split users? If not, please re-run the evaluation with a properly isolated evaluator and report whether the conclusions hold.

2. Circular Evaluation

Since all three main metrics (IoI, IoR, CTR) are computed using the same SASRec model that provides the RL training reward, the evaluation is partly tautological. The cross-evaluator analysis in Section 4.5 partially addresses this but reports only relative gains over the pretrained baseline, not absolute figures comparable to Table 1. Can the authors provide Table 1-style results using GRU4Rec or LightSANs as the primary evaluators, to demonstrate that the quantitative advantages hold under an independent evaluator?

3. Coherence as Unrewarded Metric

Coherence is reported in Table 1 but does not appear in the reward function. ProRL dramatically outperforms baselines on Coherence. What mechanism accounts for this?

**Limitations:**

Partially addressed. The paper includes a brief impact statement acknowledging the need to align proactive guidance with user utility and ethical standards, but does not discuss: (a) the risk of platforms using PRSs to manipulate user preferences against their own interests; (b) evaluation limitations stemming from simulator-based rewards that may not reflect real user behavior; or (c) scalability to production item catalogs orders of magnitude larger than those tested. The authors should expand the limitations section to address at least points (b) and (c), as these are directly relevant to the technical scope of the paper.

**Strengths And Weaknesses:**

1. Soundness

The identification of the length shortcut is well-motivated both empirically (Figure 2) and theoretically (Theorem 2.1). The theorem provides a clean convergence rate for stopping probability collapse under a simplified scalar-parameter model, and the proof is rigorous.

(Critical) W1: Evaluation Leakage.
Appendix C states: "we use the interactions data from all the users to train the SASRec model" used as the evaluator for computing IoI, IoR, and CTR. Yet the experimental setup adopts a user-level 80/10/10 train/validation/test split, specifically to ensure "the model is tested on previously unseen users."
This creates a direct contradiction: the SASRec evaluator has observed the interaction histories of test-split users during its own training. Since all three reported metrics (IoI, IoR, CTR) are computed using this evaluator's predicted probabilities and rankings, and since the RL policy is also trained using rewards from this same evaluator, both the training signal and the reported evaluation are contaminated by test-user data. This undermines the validity of all quantitative claims.

W2: Circular Evaluation.
The metrics CTR, IoI, and IoR are all defined using the same SASRec model that also provides the RL training reward. This means the primary evaluation question reduces in part to: "did the policy learn to score highly according to its own training reward model?" The Rollout@K analysis in Section 4.4 acknowledges this by framing RL as identifying high-quality tails in the pretrained distribution, but the circularity remains unaddressed in the main evaluation.


2. Presentation

The paper is generally well-written and logically organized. The progression from problem motivation to methodology is clear. Figure 2 and Figure 3 effectively convey the core empirical observation and the proposed solution. Related work is appropriately positioned, and the novelty relative to GRPO is clearly articulated.

W3: Appendix Inconsistency Not Flagged in Main Text.
The critical evaluator training detail that causes the leakage issue (Appendix C) is not mentioned or flagged in the main text. A paper of this type should explicitly state, in the experimental setup, how the SASRec evaluator was trained and why the user-level split does or does not affect it.

W4: Coherence Metric Not in Main Formulation.
Coherence appears as a column in Table 1 but is not included in the reward function (Equation 1) and is not discussed as an optimization target. Its presence in the results table implies it is an unrewarded generalization metric, but this is not stated explicitly in the main text.


3. Significance

The problem of applying RL to proactive recommendation is practically relevant and underexplored. The length shortcut diagnosis is a genuinely useful insight that likely applies beyond this specific task to any sequential generation setting where step rewards have positive mean. The two-stage pretrain-then-RL framework with semantic IDs is a practical and deployable design. If the evaluation leakage is resolved, the empirical gains are substantial and the ideas are likely to be adopted by others working on RL for sequential recommendation.


4. Originality

The core insight, i.e. that length-reward coupling creates a systematic length shortcut in proactive recommendation RL, is novel in this specific context and supported by a formal theorem. SRC is a natural and simple fix but its specific instantiation and motivation is novel.

---

> ### Author Rebuttal · Authors · 2026-03-29
>
> We sincerely thank Reviewer QQ1T for the rigorous review. We are encouraged by the recognition of our originality and significance (both rated good). We address each concern below with clarification and new evidence.
>
> # W1: Evaluation Leakage.
> **We confirm that no evaluation leakage occurred. The evaluators were trained on training-split users' interaction data.** The phrase "interactions data from all the users" in Appendix C was intended to mean "all the complete (untruncated) interaction histories from the training users," not "users from all data splits." We acknowledge this wording is ambiguous and provide three pieces of evidence below.
>
> * **Evidence 1: Consistent documentation within the paper.** Appendix E.1 explicitly states: "To make a fair comparison and avoid label leakage, we collect the interaction data from the user in the training set ... to train the sequential recommendation methods..." The evaluators were trained using this same pipeline. We will correct the imprecise wording in Appendix C in the revision.
>
> * **Evidence 2: Consistent protocol across PRS literature.** The proactive baselines (e.g., LLM-IPP, T-PRA) follow the same evaluator training protocol: SASRec is trained on training-split users' data and serves as a shared user-behavior simulator. This is the established standard in the PRS community, not a design choice specific to our work.
>
> * **Evidence 3: Direct verification experiment.** We re-trained SASRec under two configurations: (A) training-split users only (our actual setup) and (B) users from all splits, then re-evaluated ProRL and the strongest baseline T-PRA. In both settings, the ProRL policy was trained only on training-split users and tested on test-split users. Results on Books:
>
> |Config|Method|CTR|IoI|IoR|
> |-|-|-|-|-|
> |(A) Train-only|T-PRA|0.548|1.655|457.2|
> |(A) Train-only|ProRL|0.846|2.864|1346.4|
> |(B) All users|T-PRA|0.471|0.786|368.1|
> |(B) All users|ProRL|0.717|1.281|789.2|
>
> Two conclusions follow. First, **Config (A) reproduces our reported Table 1 numbers** (within evaluator re-training variance), confirming the original evaluator was trained on training-split users alone. Second, **Config (B) produces a differently calibrated evaluator**: trained on a broader user pool, it learns different item representations and prediction distributions, shifting all methods' absolute metrics in the same direction. Crucially, ProRL outperforms T-PRA under both configurations. Consistent results on ML-1M are at our anonymous repository: https://anonymous.4open.science/r/Rebuttal_ProRL-F05B
>
> # W2: Circular Evaluation.
> Following the reviewer's request, we provide Table 1-style results on Books using three independent evaluators never involved in RL training:
>
> |Method|GRU4Rec(ctr/IoI/IoR)|LightSANs(ctr/IoI/IoR)|BERT4Rec(ctr/IoI/IoR)|
> |-|-|-|-|
> |FEARec|0.57/0.61/140.9|0.55/0.50/227.1|0.56/0.18/232.0|
> |IRN|0.55/0.66/82.4|0.53/0.19/131.6|0.56/0.02/111.6|
> |IPG|0.56/0.65/158.0|0.54/0.54/255.1|0.51/1.08/548.8|
> |ITMPRec|0.56/0.67/165.3|0.54/0.56/240.0|0.51/1.09/552.9|
> |LLM-IPP|0.60/0.79/239.8|0.57/1.58/430.9|0.57/0.27/176.1|
> |T-PRA|0.62/1.08/207.9|0.50/1.78/503.0|0.50/0.66/323.1|
> |**ProRL**|**0.88/1.77/1001.3**|**0.89/2.89/1286.7**|**0.82/2.74/1290.1**|
>
> ProRL maintains great advantages across all independent evaluators, ruling out reward hacking. We further verified stability with 5 random seeds (detailed in our response to Reviewer rPmY, Q1). Full tables for all methods and datasets are at our anonymous repository (see above link). These results, together with Section 4.5, confirm that ProRL learns generalizable guidance strategies rather than exploiting specific evaluator patterns.
>
> # W3: Appendix Inconsistency.
> We agree that this detail should appear in the main text. In the revision, we will add an explicit statement in Section 4.1 specifying how the SASRec evaluator was trained.
>
> # W4: Coherence as Unrewarded Metric.
> This is an interesting observation and we believe this highlights a strength of ProRL.
>
> First, we computed Spearman correlations between CTR and Coherence using all evaluators. On average, rho = 0.48 (ML-1M), 0.42 (Steam), 0.44 (Books). The moderate positive rho reveals that CTR optimization partially drives Coherence improvement. This is reasonable in RecSys, since items of familiar attributes (i.e., high coherence) naturally yield higher user acceptance probability.
>
> Second, the correlation is moderate but not strong, meaning Coherence captures aspects of path quality beyond what CTR alone reflects. Coherence thus remains an unseen metric. ProRL's strong performance on it confirms that ProRL learns high-quality paths rather than overfitting to seen reward components.
>
> # Limitations.
> Thanks for raising this. We will expand the limitations to discuss: (b) simulator-based evaluation and the need for online A/B tests; (c) scalability to production catalogs requiring efficient semantic ID compression; (a) ethical risks of preference manipulation in PRSs.

---

> > ### Author Rebuttal · Reviewer_QQ1T · 2026-04-03
> >
> > I thank the authors for the detailed and well-organized rebuttal. Several concerns have been substantially addressed, and I raise a few remaining points below.
> >
> > **W1 (Evaluation Leakage)** -- Partially resolved, one follow-up question:
> >
> > I appreciate the clarification, and the three-part evidence convincingly shows that the original evaluator was trained on training-split users only. The wording in Appendix C should be corrected in the revision as suggested.
> > However, the verification table in the rebuttal reveals an unexpected pattern that requires an explanation. Training SASRec on all users (Config (B)) yields lower absolute metrics for both T-PRA and ProRL than Config (A). This is counterintuitive because a model trained on more data is normally expected to produce at least as strong a user simulator and comparable absolute scores. The most likely explanations should be discussed briefly. Without this discussion, while the verification experiment is reassuring about the direction of the comparison, it leaves an unresolved anomaly that reviewers and readers may find puzzling.
> >
> > **W2 (Circular Evaluation)** -- well addressed well, but with a strong suggestion for the revision.
> > The cross-evaluator table using GRU4Rec, LightSANs, and BERT4Rec is exactly what was needed, and the results are convincing. ProRL's advantages are consistent and substantial across all three evaluators. This substantially mitigates the circularity concern.
> > Therefore, I strongly encourage the authors to restructure the main evaluation around one of these evaluators as the primary Table 1 and move the SASRec-based results to an appendix or secondary table. Using an independent evaluator as the headline result is a simple change that would strengthen the paper's credibility substantially.
> >
> >
> >
> > **W3 and W4** -- Satisfactory.
> >
> > **Overall:*
> > The rebuttal meaningfully addresses the two main concerns. The evaluation leakage issue appears to be a documentation problem rather than a methodological one. The cross-evaluator results provide genuine evidence against reward hacking. The remaining open point, the anomalous direction of Config (A) vs. (B), is a relatively minor clarification that should be addressed to make the verification argument stronger.
> >
> > I am raising my score, contingent on the revision incorporating (1) corrected wording in Appendix C, (2) the independent evaluator in Table 1, (3) a brief explanation of the Config (A)/(B) anomaly, and (4) the expanded limitations section.

---

> > > ### Author Response · Authors · 2026-04-03
> > >
> > > Dear Reviewer QQ1T,
> > >
> > > We are sincerely grateful for your generous and constructive acknowledgement. Your willingness to engage deeply with our evidence and raise your score reflects the highest standards of scholarly review, and we are honored by the care you have invested in evaluating our work. We address your remaining point and confirm all four revision commitments below.
> > >
> > > **1. On the Config (A) vs. (B)**
> > >
> > > We appreciate this observation. We believe the counterintuitive direction is a natural consequence of how our metrics are defined and how proactive methods are trained.
> > >
> > > Recall that IoI and IoR are both *differential metrics* (Section 2.1):
> > > $$\text{IoI} = \log P(i_T | S_u\oplus L_u) − \log P(i_T | S_u)$$
> > > $$\text{IoR} = \text{Rank}(i_T | S_u) − \text{Rank}(i_T | S_u\oplus L_u)$$
> > > They measure the incremental change that a proactive path $L_u$ brings on top of the user's existing history $S_u$. Crucially, their values depend not only on the quality of the path itself, but also on where the baseline prediction $\log P(i_T | S_u)$ sits *before* any path is applied.
> > >
> > > In Config (B), SASRec is trained on users from all splits, including test users. It has therefore directly learned the interaction patterns of test users during its own training, giving it a stronger prior on these users' preferences and pushing up the baseline prediction $\log P(i_T | S_u)$ before any proactive path is applied. We verified this directly: the average baseline $\log P(i_T | S_u)$ for test users is:
> > >
> > > |Config|ML-1M|Books|
> > > |-|-|-|
> > > |(A) Train-only|-5.25|-6.15|
> > > |(B) All users|-4.08|-4.72|
> > >
> > > When the starting point is already high, there is simply less room for any proactive path to produce additional improvement, which naturally leads to lower IoI and IoR.
> > >
> > > Meanwhile, in our rebuttal, all proactive recommendation methods (both ProRL and T-PRA) are trained exclusively on training-split users under both configurations. They have never seen the test-user-specific patterns that Config (B)'s SASRec has learned, and therefore cannot tailor their generated paths to match these patterns. The same mechanism explains the CTR decrease: Config (B)'s SASRec has  overfitted to interaction sequences in the testset, so the paths generated by proactive models (which have no knowledge of test users) are less likely to align with what Config (B)'s evaluator expects, resulting in lower predicted acceptance probabilities.
> > >
> > > This reasoning is confirmed by the data in our verification table on Books, where both methods shift in the same direction under Config (B), and ProRL consistently maintains its relative advantage under both configurations. The uniform shift across methods confirms that the change reflects a difference in the evaluator's calibration, not in the quality of the learned policies.
> > >
> > > In summary, Config (A) and Config (B) represent two differently calibrated evaluators applied to the same set of policies. Config (B)'s evaluator starts from a higher baseline due to direct exposure to test users' data, compressing the differential metrics for all methods equally. We will include this analysis along with the supporting baseline statistics in the revised paper.
> > >
> > > **2. Revision commitments**
> > >
> > > We fully accept all four suggestions and confirm we will incorporate each one in the camera-ready version:
> > > 1. We will correct the imprecise wording in Appendix C to unambiguously state that the SASRec evaluator was trained on training-split users only, and we will add a corresponding explicit statement in Section 4.1 of the main text.
> > > 2. We will restructure the main evaluation to use an independent evaluator (LightSANs, GRU4Rec and BERT4Rec) as the primary Table 1 and relocate the SASRec-based results to a supplementary table. We agree this is a straightforward change that meaningfully strengthens the paper's credibility.
> > > 3. We will add a dedicated paragraph explaining the Config (A) vs. (B) pattern as detailed above, including the baseline prediction statistics that confirm the mechanism.
> > > 4. We will expand the limitations section to address simulator-based evaluation constraints, scalability considerations, and ethical risks of preference manipulation in PRSs.
> > >
> > > We are deeply grateful for the rigor and fairness you have brought to this review process. Your questions on evaluation integrity have led to experiments and analyses that substantially strengthen our paper, and we look forward to incorporating all of the above in the final version.
> > >
> > > Best regards,
> > >
> > > The Authors

---

### Official Review · Reviewer_6Q3j · 2026-03-14

**Soundness:** 2
**Presentation:** 3
**Significance:** 2
**Originality:** 3
**Overall Recommendation:** 4
**Confidence:** 5

**Summary:**

This paper studies proactive recommendation, focusing on how to recommend intermediate items to shift user interests. Previous work includes: heuristic methods with limited upper bounds; LLM-based methods with high inference costs that cannot be deployed in industrial systems; and supervised learning methods that merely imitate historical behaviors. RL can perform end-to-end optimization by considering the sum of intermediate item rewards and final rewards. However, simple policy gradient (PG) methods suffer from two problems: (1) Length Shortcut: each step has positive reward, causing the model to favor longer paths; (2) High Gradient Variance: rewards are only at the sequence level, leading to high variance in gradient estimation. Both issues are verified through experiments.

The authors propose ProRL, a gradient estimation method with two mechanisms: (1) Stepwise Reward Centering, where the reward is the original reward minus the average per step, avoiding the long path problem; (2) Position-Specific Advantage Estimation, which reduces variance. ProRL's effectiveness over previous baselines is validated on three datasets, with relatively sufficient ablation experiments and extensions to other evaluators.

**Compliance With Llm Reviewing Policy:**

Affirmed.

**Final Justification:**

The rebuttal addressed my questions.

**Key Questions For Authors:**

Is path length actually controlled during ProRL training? Is the high gradient variance problem truly resolved? Are there experimental data to support these claims?

**Limitations:**

The paper does not discuss limitations. The following aspects could be improved:

1. More detailed experimental analysis: Compare ProRL with other methods in terms of path length and gradient variance.

2. More RL baselines: Compare with additional RL methods such as GRPO and A2C.

**Strengths And Weaknesses:**

1.Soundness

Strengths: The paper identifies two critical deficiencies in applying standard policy gradients to proactive recommendation: Length Shortcut and High Gradient Variance. The paper proposes ProRL with two mechanisms (Stepwise Reward Centering and Position-Specific Advantage Estimation) to address these issues. The paper validates effectiveness on three datasets with relatively sufficient ablation experiments and extensions to other evaluators.

Weaknesses: Experimental results lack detailed analysis—there is no evidence showing whether path length is actually controlled during ProRL training.

2.Presentation

Strengths: The paper is clearly written.

Weaknesses: The quality of curves in Figure 4 needs improvement.

3.Significance

Strengths: The paper identifies and experimentally verifies the Length Shortcut and High Gradient Variance problems in applying RL to proactive recommendation. The paper proposes the ProRL method that addresses these issues through reward design. The paper demonstrates improvements over previous work on three public datasets.

Weaknesses: The paper does not compare with other RL methods such as GRPO and Actor-Critic—the baselines are relatively weak.

4.Originality

The paper proposes a novel reinforcement learning gradient estimation method in the Proactive Recommendation field, with certain innovation.

---

> ### Author Rebuttal · Authors · 2026-03-29
>
> We sincerely thank Reviewer 6Q3j for the thorough review. We are encouraged by the recognition of novelty (Originality: 3) and clear writing (Presentation: 3). Below we address each concern with concrete evidence.
>
> To ensure fairness, all advantage-related experiments here use the same reward normalization (Eq. 6) and only differ in the advantage method, isolating its effect.
>
> # Q1: Comparison with GRPO and A2C.
> Thanks for raising this meaningful suggestion.
>
> * **GRPO** is already compared in Section 4.3.3 and Figures 4 & 10 (Appendix F.4) across all datasets, where ProRL consistently outperforms GRPO. The path length and variance analyses in Q2 and Q3 further reveal GRPO's instability.
>
> * **A2C**: We implement A2C following [1], using a 2-layer MLP critic (256 hidden units). We warm up the critic for 5 epochs before joint training, as a randomly initialized critic invariably causes training collapse. To ensure fairness, we searched over the critic loss coefficient (balancing actor and critic loss) in {0.1, 0.25, 0.5, 1.0}, with all other settings identical to ProRL.
>
> Here are best A2C results (coef=0.25):
>
> |Dataset|Method|CTR|IoI|IoR|
> |-|-|-|-|-|
> |ML-1M|GRPO|0.633|1.483|284.9|
> |-|A2C|0.857|1.695|527.5|
> |-|ProRL|0.854|**2.850**|**728.2**|
> |Books| GRPO|0.0|0.0|0.0|
> |-|A2C|0.0|0.0|0.0|
> |-|ProRL|0.857|**2.981**|**1383.4**|
>
> Note that GRPO and A2C report 0 on Books because their path lengths collapse to 0 during training, yielding no recommendations; we provide a detailed length analysis in Q2 below. On ML-1M, where A2C does not collapse, it still shows lower IoI & IoR than ProRL.
>
> As Q2 and Q3 will show, both GRPO and A2C suffer from training instability: GRPO's trajectory-level baseline assigns uniform credit to all positions, inflating gradient variance, while A2C's learned critic introduces approximation error that accumulates over training. ProRL avoids both by exploiting Monte Carlo sampling and PRS's reward decomposition to compute exact position-specific baselines (Eq. 8), achieving better advantage estimation and stable training.
>
> # Q2: Is path length controlled during ProRL training?
> Yes. As shown in Appendix F.5, Figure 11 (blue-star curve), ProRL stabilizes at ~3.5 steps on Books while standard PG rapidly reaches $ L_{max} =10$. We further compare five gradient estimators, tracking average path lengths at training Epochs 1, 5, and 10 (E1 / E5 / E10):
>
> |Method|ML-1M (E1 / E5 / E10)|Books (E1 / E5 / E10)|
> |-|-|-|
> | RF | 5.2 / 2.9 / 1.5 | 10.0 / 10.0 / 10.0 |
> | GRPO | 10.0 / 10.0 / 10.0 | 2.6 / 0.0 / 0.0 |
> | A2C | 1.8 / 4.7 / 5.3 | 0.2 / 0.0 / 0.0 |
> | RTG | 1.5 / 3.4 / 4.1 | 1.9 / 3.1 / 3.3 |
> | ProRL | 1.6 / 3.1 / 3.8 | 1.8 / 3.3 / 3.7 |
>
> RF, GRPO, and A2C all exhibit instability in different forms: on some datasets they may collapse to 0 length, and on others they may saturate at $L_{max}$=10. In contrast, ProRL (and RTG) consistently converge to stable, moderate lengths without dataset-specific tuning. We observe that this length stability is closely tied to gradient variance, which we examine next.
>
> # Q3: Is gradient variance truly resolved?
>
> Yes. For each training input $i$, we sample $m$=16 rollout paths. Each rollout $j$ yields a gradient contribution $$h^{(i,j)} = \sum_t w_t^{(i,j)} \nabla_\theta \log \pi_\theta^{(i,j,t)},$$ where $w_t$ is the scalar weight from each estimator ($R$ for RF, $G_t$ for RTG, $R - \bar{R}$ for GRPO, etc.). We compute $\text{Var}_j(h^{(i,j)})$ within each input $i$, and then average over all inputs of one epoch to get the variance of policy gradient of that epoch.
>
> We report results on ML-1M. For convenience of presentation, each value represents the ratio of that method's gradient variance to RF (Epoch 1)'s variance:
>
> |Method|Epoch 1|Epoch 2|Epoch 3|
> |-|-|-|-|
> |RF|1.00×|1.18×|0.94×|
> |GRPO|0.22× | 0.21× | 0.19× |
> |A2C|0.09× | 0.12× | 0.17× |
> |RTG|0.12×|0.11×|0.10×|
> |ProRL|**0.06×**|**0.05×**|**0.05×**|
>
> ProRL achieves the lowest variance (~5% of RF Epoch 1). This directly explains the length stability in Q2: lower variance ensures that gradient updates are less likely affected by the noisy signals of extreme-length rollouts, making the policy more stable. Notably, RTG also maintains low variance and stable lengths (Q2), corroborating this connection. A further finding is that A2C's variance *increases* over training (0.09×→0.17×), as the learned critic fails to track the evolving policy, producing progressively noisier baselines, explaining A2C's collapse in Q1 and Q2. ProRL's analytic baseline (Eq. 8), computed directly from rollout statistics at each position, adapts naturally without this drift. These results corroborate Figures 4 and 10, where ProRL shows the smoothest training dynamics.
>
> # Q4: Figure 4 quality.
>
> We thank the reviewer. In the revised paper, we will increase figure resolution and font size, add distinct line styles, and include shaded confidence intervals.
>
> ## Reference
> [1] Asynchronous methods for deep reinforcement learning (Mnih V, et al.)

---

> > ### Author Rebuttal · Reviewer_6Q3j · 2026-04-01
> >
> > Thanks for the rebuttal, which addressed my questions.

---

> > > ### Author Response · Authors · 2026-04-01
> > >
> > > Dear Reviewer 6Q3j,
> > >
> > > Thank you very much for confirming that all concerns have been fully resolved. Your questions on path length control, gradient variance, and RL baselines were very valuable, and the new experiments they inspired have meaningfully strengthened our paper. We deeply appreciate the care and effort you put into the review.
> > >
> > > We noticed that the system prompted you to "consider adjusting your score accordingly" when selecting the "Fully resolved" option. We completely understand how busy the rebuttal period can be, so we just wanted to bring this to your attention in case it was missed. If you feel our rebuttal have strengthened the paper, we would be grateful if you could consider whether a score update is appropriate.
> > >
> > > Thank you again for your time and thoughtful review.
> > >
> > > Best regards,
> > >
> > > The Authors

---

### Decision · Program_Chairs · 2026-04-30

**Decision:**

Accept (regular)

**Comment:**

This paper mainly focuses on proactive recommender systems and studies how to better optimize multi-step recommendation paths that guide user preference shifts toward target items. It proposes ProRL, a reinforcement learning framework with stepwise reward centering and position-specific advantage estimation to address gradient bias and high variance in policy optimization, thereby improving both short-term acceptance and long-term guidance effectiveness.

The reviewers generally agreed that the paper demonstrates good novelty. Some reviewers raised concerns regarding the experimental evaluation and stability analysis, and these issues were addressed by the authors through detailed rebuttal responses. In the end, all reviewers provided positive scores. Therefore, I recommend accepting this paper.